# CARPRT: CLASS-AWARE ZERO-SHOT PROMPT REWEIGHTING FOR BLACK-BOX VISION-LANGUAGE MODELS

**Ruijiang Dong** [1]*, **Zesheng Ye**[1]*, **Jianzhong Qi**[1], **Lei Feng**[2,3], **Feng Liu**[1,3]†
**Gang Niu**[3], **Masashi Sugiyama**[3,4]
[1]University of Melbourne, [2]Southeast University,
[3]RIKEN Center for Advanced Intelligence Project, [4]The University of Tokyo
`ruijdong@student.unimelb.edu.au, fenglei@seu.edu.cn`
`{zesheng.ye, jianzhong.qi, feng.liu1}@unimelb.edu.au`
`gang.niu.ml@gmail.com, sugi@k.u-tokyo.ac.jp`

## ABSTRACT

Pre-trained *vision-language models* (VLMs) enable zero-shot image classification by computing the similarity score between an image and textual descriptions, typically formed by inserting a class label (e.g., "cat") into a prompt (e.g., "a photo of a"). Since the score for a given image-class pair is sensitive to the choice of prompt, existing studies ensemble multiple prompts using a *weighting vector* to aggregate scores across different prompts. Yet, in current strategies, the weighting vector assigned to each prompt is *shared* across all classes, implicitly assuming that prompts are conditionally independent of classes, which often does not hold in practice, as a prompt like "an aerial view of" might be apt for "airport" but ill-suited for "apple". To address this, we propose *class-aware zero-shot prompt reweighting* (CARPRT). This scoring scheme adjusts the weighting vector for each class label by capturing the class-specific relevance of different prompts in a *training-free* manner. For each class label and every available prompt, we quantify their class-specific relevance by averaging image–text relevance scores over images predicted to that class under the given prompt. These estimates are then normalized to derive class-specific weights. Evaluations on standard image classification benchmarks show that CARPRT outperforms existing class-independent reweighting methods, confirming that modeling prompt-class dependencies is crucial for effective zero-shot prediction and even broader VLM-based application settings that rely on prompt ensembling. Our code is available at https://github.com/tmlr-group/CARPRT.

# 1 INTRODUCTION

*Vision-language models* (VLMs) have transformed how machine learning models interpret visual content by jointly leveraging visual and textual modalities. Models like CLIP (Radford et al., 2021) and DeCLIP (Li et al., 2022) enable *zero-shot image classification* by computing similarity scores between an image and textual descriptions of class labels, then predicting the label with the highest score. By forming textual descriptions of labels (e.g., *"a photo of a [label]"*), this approach—known as *prompting*—removes the need for task-specific training to recognize visual concepts.

However, these models' *zero-shot performance* is sensitive to the precise wording of prompts, as subtle phrasing changes can significantly alter the perceived relevance of visual features, leading to different similarity scores and classification outcomes (Radford et al., 2021). Identifying phrasings that remain effective across diverse visual concepts is challenging and often yields inconsistent results across datasets (Allingham et al., 2023). This sensitivity means that manually crafting optimal prompts for each class or dataset, while helpful for performance, becomes laborious and unreliable in large-scale settings. Recent work has explored using *large language models* (LLMs) to generate

---

*Equal contribution.
†Corresponding Author.

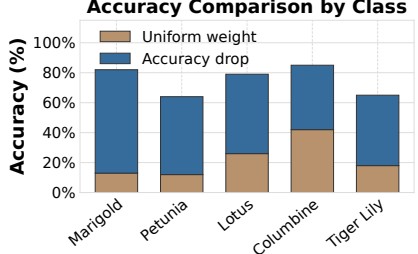
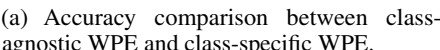
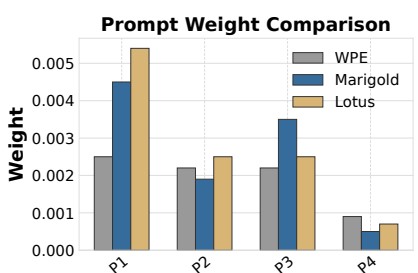

(a) Accuracy comparison between class-agnostic WPE and class-specific WPE.

(b) Optimal weight comparison between class-specific WPE and class-agnostic WPE.

Figure 1: Empirical motivation for class-specific weighting on Flower102 (Nilsback & Zisserman, 2008). We showcase the results of five classes by shifting from class-agnostic WPE to class-specific WPE (using ground-truth labels), and the estimated optimal weights under two weighting schemes, confirming that optimal prompt weights are class-dependent.

richer class descriptions, but this introduces heavy computational overhead, reducing the efficiency that makes zero-shot methods attractive in the first place.

This paper focuses on a more prevalent question: improving zero-shot classification when only a *fixed* set of *predefined prompts* and *unlabeled images* are available at inference, *under black-box access*, which requires methods that leverage only the inference data to *optimize prompt utilization*. A common strategy is *prompt ensembling*, which averages embeddings of multiple prompts to produce stable class representations (Radford et al., 2021). However, this approach assumes equal prompt contributions—a simplification that harms downstream performance when misaligned templates are included. Allingham et al. (2023) advanced this concept by determining prompt-specific weights using unlabeled data, depending on how compatible each prompt is with the downstream task. This method achieves results comparable to manually selected templates. Still, while such methods vary weights across prompts, they assign *the same weight across all classes* to each prompt.

We argue that this class-agnostic reweighting is suboptimal. Intuitively, different semantic classes vary in their affinity to different prompts. For example, a prompt like "*This is a photo of a [label], a type of fruit*" is more relevant to class "*strawberry*," but ill-suited for class "*lamb*", which would better match "*This is a photo of a [label], a type of animal*" instead. This implies that optimal prompt utilization may require class-specific considerations. To validate this intuition, we conduct controlled proof-of-concept experiments on the Flower102 dataset (Nilsback & Zisserman, 2008) (Fig. 1). By applying Weighted Prompt Ensembling (WPE) (Allingham et al., 2023) *independently* to images of each class (thus simulating "perfect" class-specific knowledge for weight estimation), we observe accuracy gains compared to global WPE that estimates a single set of class-agnostic weights (Fig. 1(a)). Moreover, the optimal prompt weights vary across classes[1] (Fig. 1(b)), rather than being globally shared.

We further study this observation theoretically and present a probabilistic framework (Sec. 3) to clarify the underlying mechanism of prompt ensembling. We show that class-agnostic weighting schemes, such as WPE, indeed implicitly assume conditional independence between the class label and the prompt weights given an image. This assumption, however, may not always reflect real-world data characteristics and limit the expressivity of such weighting schemes as a result.

Building on insights, we introduce **Class-Aware Zero-shot Prompt ReweighTing** (CARPRT), a *training-free* method to infer class-specific prompt weights using only unlabeled images. Unlike our controlled proof-of-concept experiment, CARPRT does *not* require ground-truth labels for weight estimation. Instead (Sec. 4), for each image, CARPRT first calculates similarity scores against all possible prompt-class combinations using a pre-trained VLM (e.g., CLIP (Radford et al., 2021)) via *forward inference only*. It then assigns a pseudo-class label to the image based on the combination yielding the highest score. These pseudo-labels are then used to aggregate information for class-specific weight derivation: for each class, the weight for a given prompt is determined by the maximum similarity that prompt achieves in conjunction with that (pseudo-)class across the reference images. This scheme helps tailor the prompt ensemble to the semantic content of each category.

---

[1]The prompt templates denoted in Fig. 1(b) are: P1 = "a photo of a , a type of flower.", P2 = "satellite photo of .", P3 = "a close-up photo of the .", P4 = "a drawing of a ."

We empirically evaluate CARPRT on ten fine-grained *zero-shot classification* benchmarks (Sec. 5), ImageNet (Russakovsky et al., 2015) (and its variants), and explore its utility in broader VLM-based adaptation scenarios such as prompt tuning (App. G). Our results show that CARPRT consistently outperforms existing prompt ensembling/reweighting schemes across VLM architectures and backbones, highlighting that incorporating class-awareness is a way to maximize the potential of prompt ensembling for zero-shot classification, with potential benefits for a wide range of VLM applications.

## 2 PROBLEM SETTING AND RELATED WORK

**Zero-Shot Prediction with VLM**. VLMs such as CLIP (Radford et al., 2021) achieve visual-text alignment through large-scale contrastive pre-training. It consists of an image encoder $f : \mathcal{X} \to \mathcal{Z}$ and a text encoder $g : \mathcal{T} \to \mathcal{Z}$, mapping images from space $\mathcal{X}$ and texts from space $\mathcal{Y}$ into a shared embedding space $\mathcal{Z}$. The alignment is driven by maximizing the cosine similarity between the embeddings of matched image-text pairs while minimizing it for non-matched pairs.

This alignment enables *zero-shot image classification*. For a set of $C$ classes $\mathcal{Y} = \{y_1, \ldots, y_C\}$, each class $y_c$ is mapped to a text description $\boldsymbol{t}_c$ via a prompt template $p : \mathcal{Y} \to \mathcal{T}$, such as $\boldsymbol{t}_c =$"A photo of $\{y_c\}$.". The text encoder $g(\cdot)$ then produces class embeddings $\boldsymbol{z}^{\mathrm{T}} = [\boldsymbol{z}_1^{\mathrm{T}} \ \boldsymbol{z}_2^{\mathrm{T}} \ \cdots \ \boldsymbol{z}_C^{\mathrm{T}}]^{\top}$ where $\boldsymbol{z}_c^{\mathrm{T}} = g(\boldsymbol{t}_c)$ for $c \in \{1, \ldots, C\}$. Given an image $\boldsymbol{x} \in \mathcal{X}$ with its embedding $\boldsymbol{z}^{\mathrm{I}} = f(\boldsymbol{x})$, the predicted class is given by $\hat{y} = \arg\max_{c \in \{1, \ldots, C\}} \text{sim}\left(\boldsymbol{z}^{\mathrm{I}}, \boldsymbol{z}_c^{\mathrm{T}}\right)$, i.e., one whose text embedding $\boldsymbol{z}_c^{\mathrm{T}}$ has the highest cosine similarity with $\boldsymbol{z}^{\mathrm{I}}$. This allows for zero-shot classification based on semantic alignment without task-specific fine-tuning. Yet, the classification performance is highly sensitive to the choice of prompt template $p$. An ill-suited template can lead to misaligned class embeddings.

This work focuses on mitigating this sensitivity by ensembling *multiple **predefined** templates* $\mathbb{P} = \{p_1, \ldots, p_n\}$, particularly when $\mathbb{P}$ is fixed, without relying on additional labeled data. That is, in the *zero-shot classification* setting, we consider the following problem[2]:

**Problem 1** (Prompt Ensembling)**.** *Given a pre-trained VLM with an image encoder $f$ and a text encoder $g$, a label space $\mathcal{Y}$ with $C$ classes, a fixed prompt template set $\mathbb{P}$ with $|\mathbb{P}| = n$, and an unlabeled image dataset $\mathbb{D} = \{\boldsymbol{x}_1, \ldots, \boldsymbol{x}_m\}$, construct the class embeddings $\boldsymbol{z}^{\mathrm{T}}$ using a prompt weight matrix $\mathbf{W} \in \mathbb{R}^{n \times C}$, where each row $\mathbf{W}_c = [w_{1,c}, \ldots, w_{n,c}]^{\top}$ refers to weights of $n$ prompts for class $y_c \in \mathcal{Y}$, subject to $w_{i,c} \geq 0$ and $\sum_{i=1}^{n} w_{i,\cdot} = 1$. The text embeddings for class $y_c$ are thus*

$$\begin{bmatrix} \boldsymbol{z}_1^{\mathrm{T}} \\ \vdots \\ \boldsymbol{z}_C^{\mathrm{T}} \end{bmatrix} = \frac{1}{n} \left( \begin{bmatrix} \boldsymbol{z}_{1,1}^{\mathrm{T}} & \boldsymbol{z}_{2,1}^{\mathrm{T}} & \cdots & \boldsymbol{z}_{n,1}^{\mathrm{T}} \\ \vdots & \vdots & \ddots & \vdots \\ \boldsymbol{z}_{1,C}^{\mathrm{T}} & \boldsymbol{z}_{2,C}^{\mathrm{T}} & \cdots & \boldsymbol{z}_{n,C}^{\mathrm{T}} \end{bmatrix} \cdot \begin{bmatrix} w_{1,1} & w_{1,2} & \cdots & w_{1,C} \\ \vdots & \vdots & \ddots & \vdots \\ w_{n,1} & w_{n,2} & \cdots & w_{n,C} \end{bmatrix} \right). \tag{1}$$

*where $\boldsymbol{z}_{i,c}^{\mathrm{T}} = g(p_i(y_c))$ is the text embedding for class $y_c$ under prompt $p_i$. The objective is then to find the set of all such weight vectors $\mathbf{W} = \{\mathbf{W}_c\}_{c=1}^{C}$ that would (ideally) minimize the empirical zero-shot classification error over the unlabeled dataset $\mathbb{D}$, i.e., correctly predict the (unknown) ground-truth label $y_j$ by $\hat{y}_j$ for each $\boldsymbol{x}_j \in \mathbb{D}$.*

Existing *prompt ensembling* schemes can be viewed as constrained versions of the general formulation in Problem 1, differing primarily in how they determine the prompt weights $\mathbf{W}$.

**Mean Prompt Ensembling (MPE) as a Solution.** The most straightforward approach, MPE (Radford et al., 2021), averages text embeddings from multiple prompts, equivalently setting $w_{i,c} = 1$ for all prompts $p_i$ and classes $y_c$ in Eq. 1, such that $\mathbf{W}$ reduces to an *all-ones* matrix. MPE seeks to improve robustness over single-prompt usage by diversifying textual inputs. Yet, treating all prompts equally can impair the efficacy if $\mathbb{P}$ is semantically misaligned with the downstream task $\mathbb{D}$.

**Weighted Prompt Ensembling (WPE) as a Solution.** To mitigate the impact of task-irrelevant prompts, WPE (Allingham et al., 2023) (originally termed ZPE) extends MPE by assigning data-driven weights to the prompts. WPE assesses whether a prompt $p_i$ yields generally high similarity scores over all classes with samples of $\mathbb{D}$, and up-weights more relevant ones. Each prompt $p_i$ is

---

[2]We note that there are some VLM adaptation settings, e.g., prompt tuning (Zhou et al., 2022a;b), which are not the focus of this work. To clarify, App. B details the relationship between Problem 1 and other settings.

assigned a weight via $w_{i,:} = \frac{1}{m} \sum_{j=1}^{m} \max_{c \in \{1,\ldots,C\}} \mathrm{sim}(\boldsymbol{z}_j^{\mathrm{I}}, \boldsymbol{z}_{i,c}^{\mathrm{T}})$, which, after normalization, is applied uniformly across classes $w_{i,1} = w_{i,2} = \cdots = w_{i,C}$. While WPE can down-weight unhelpful prompts, it still assumes: a prompt deemed useful (or not) is considered so for all classes *equally*.

**Can We Bridge the Gap?** As Fig. 1 shows, a prompt's efficacy often depends on the specific class it describes. Both MPE and WPE largely *neglect* this class-prompt interaction, nor attempt to understand *why* class specificity is necessary to determine prompt relevance and *how* statistical tools help to address it. To bridge this gap, we next present a probabilistic framework, establishing a principled connection between *class-aware prompt reweighting* and *zero-shot classification*.

## 3 UNDERSTANDING PROMPT REWEIGHTING: A PROBABILISTIC VIEWPOINT

Zero-shot classification with VLMs can be framed as estimating the conditional probability $\Pr(y^*|\boldsymbol{x}^*, \mathbb{P}, \mathbb{D})$ of a label $y^*$ given a query image $\boldsymbol{x}^*$, a set of prompts $\mathbb{P}$, and an unlabeled dataset $\mathbb{D}$. To understand how prompt reweighting influences this process, we develop a probabilistic framework that reveals why class-aware reweighting is necessary.

Let $\mathbf{W} \in \mathcal{W}$ be a weight matrix. We begin by marginalizing over the weight space $\mathcal{W}$ as

$$\Pr(y^*|\boldsymbol{x}^*, \mathbb{P}, \mathbb{D}) = \int_{\mathcal{W}} \Pr(y^*|\boldsymbol{x}^*, \mathbb{P}, \mathbb{D}, \mathbf{W}) \Pr(\mathbf{W}|\boldsymbol{x}^*, \mathbb{P}, \mathbb{D}) \mathrm{d}\mathbf{W}, \tag{2}$$

where $\Pr(\mathbf{W}|\boldsymbol{x}^*, \mathbb{P}, \mathbb{D})$ can further simplify to $\Pr(\mathbf{W}|\mathbb{P}, \mathbb{D})$, since in zero-shot settings, $\mathbf{W}$ is determined before access to the new query image $\boldsymbol{x}^*$. This decomposition suggests two essential tasks in zero-shot classification: (i) modeling prompt weights $\Pr(\mathbf{W}|\mathbb{P}, \mathbb{D})$ and (ii) making aggregated predictions $\Pr(y^*|\boldsymbol{x}^*, \mathbb{P}, \mathbb{D}, \mathbf{W})$ weighted by $\Pr(\mathbf{W}|\mathbb{P}, \mathbb{D})$. As such, we will continue to explore how further expansions can *inform and align with practical implementations*.

**Modeling Weight $\Pr(\mathbf{W}|\mathbb{P}, \mathbb{D})$.** Using Bayes' theorem and considering $m$ i.i.d. samples $\boldsymbol{x}_j \in \mathbb{D}$,

$$\Pr(\mathbf{W}|\mathbb{P}, \mathbb{D}) \propto \Pr(\mathbf{W}|\mathbb{P}) \Pr(\mathbb{D}|\mathbf{W}, \mathbb{P}) = \Pr(\mathbf{W}|\mathbb{P}) \prod_{j=1}^{m} \Pr(\boldsymbol{x}_j|\mathbf{W}, \mathbb{P}), \tag{3}$$

where $\Pr(\mathbf{W}|\mathbb{P})$ is the prior over weights (details are deferred to App. H) and the data (image) likelihood $\Pr(\boldsymbol{x}_j|\mathbf{W}, \mathbb{P})$ is obtained by marginalizing over classes $y_c \in \mathcal{Y}$ further:

$$\Pr(\boldsymbol{x}_j|\mathbf{W}, \mathbb{P}) = \sum_{y_c \in \mathcal{Y}} \Pr(\boldsymbol{x}_j|y_c, \mathbf{W}, \mathbb{P}) \Pr(y_c|\mathbf{W}, \mathbb{P}), \tag{4}$$

which describes how it depends on class priors and class-conditional likelihood.

**Modeling Class Prior $\Pr(y_c|\mathbf{W}, \mathbb{P})$.** For zero-shot classification where $\mathbb{D}$ is large enough, the class prior $\Pr(y_c|\mathbf{W}, \mathbb{P})$ can be estimated from pseudo-labels (i.e., predictions from a pre-trained VLM).

**Proposition 1.** *Let $\mathbb{D} = \{\boldsymbol{x}_j\}_{j=1}^{m}$ be an unlabeled dataset with unobserved classes $\mathcal{Y} = \{y_c\}_{c=1}^{C}$, and $\Pr(y_c)$ be the true class probability for class $y_c$. As $m$ grows, the empirical class distribution $\widehat{\Pr}(y_c|\mathbf{W}, \mathbb{P})$ from pseudo-labels converges to $\Pr(y_c)$ with exponentially decreasing error probability. Specifically, for any $\epsilon > 0$, we have: $\Pr\{|\widehat{\Pr}(y_c|\mathbf{W}, \mathbb{P}) - \Pr(y_c)| \geq \epsilon\} \leq 2\exp(-2m\epsilon^2)$. This implies that we can approximate true distributions by*

$$\widehat{\Pr}(y_c|\mathbf{W}, \mathbb{P}) = \frac{n_c}{\sum_{y_{c'} \in \mathcal{Y}} n_{c'}}, \ \forall y_c \in \mathcal{Y}, \tag{5}$$

*where $n_c = \sum_{j=1}^{m} \mathbb{1}_{\hat{y}_j = y_c}$ counts the images pseudo-labeled as class $y_c$ over all samples in $\mathbb{D}$.*

**Modeling Likelihood $\Pr(\boldsymbol{x}_j|y_c, \mathbf{W}, \mathbb{P})$.** Given that images $\boldsymbol{x}_j$ often lie in high-dimensional spaces, directly modeling the class-conditional likelihood can be challenging. We therefore adopt Energy-based Models (EBMs) (LeCun et al., 2006) that excel at modeling high-dimensional distributions by defining an *unnormalized* energy function, normalized by a partition function. Interpreting $\mathrm{sim}(\boldsymbol{z}_j^{\mathrm{I}}, \boldsymbol{z}_c^{\mathrm{T}})$ as the negative energy (lower energy means more likely), we have

$$\Pr(\boldsymbol{x}_j|y_c, \mathbf{W}, \mathbb{P}) = \frac{1}{Z(y_c, \mathbf{W}, \mathbb{P})} \exp\left\{\mathrm{sim}(\boldsymbol{z}_j^{\mathrm{I}}, \boldsymbol{z}_c^{\mathrm{T}})\right\}, \tag{6}$$

where $\boldsymbol{z}_j^{\mathrm{I}} = f(\boldsymbol{x}_j)$ is the image embedding, $\boldsymbol{z}_c^{\mathrm{T}} = g(p_i(y_c))$ is weighted text embedding for class $y_c$ using $\mathbf{W}_c$ (from $\mathbf{W}$). While the partition function $Z(y_c, \mathbf{W}, \mathbb{P}) = \int_{\mathcal{X}} \exp(\mathrm{sim}(\boldsymbol{z}^{\mathrm{I}}, \boldsymbol{z}_c^{\mathrm{T}}))\mathrm{d}\boldsymbol{x}$ makes exact computation intractable, for classification we only need relative likelihoods of different classes.

**Lemma 1** (Relative Likelihood). *Assume* $\mathrm{sim}(\boldsymbol{a}, \boldsymbol{b}) = \boldsymbol{a}^\top \boldsymbol{b}$ *(for $\ell_2$-normalized embeddings), then:*

$$\Pr(\boldsymbol{x}_j | y_c, \mathbf{W}, \mathbb{P}) \propto \exp\left\{\mathrm{sim}(\boldsymbol{z}_j^{\mathrm{I}}, \boldsymbol{z}_c^{\mathrm{T}})\right\} \propto \exp\left\{\sum_{i=1}^{n} (w_{i,c}\,\boldsymbol{z}_{i,c}^{\mathrm{T}})^\top \cdot \boldsymbol{z}^{\mathrm{I}}\right\}. \tag{7}$$

*This proportion relationship shows that class-specific weights $w_{i,c}$ (for $c \in \{1, \ldots, C\}$) indeed determine the influence of each prompt $p_i$ (via its embedding $\boldsymbol{z}_{i,c}^{\mathrm{T}}$) on the likelihood for class $y_c$.*

**Why Class-Specific Weighting Matters**. It is easy to check that Lemma 1 (proof in App. I) aligns with the most *general form* of prompt ensembling (Eq. 1). Crucially, class-agnostic weighting (i.e., independent) schemes, such as WPE, *deviate from this form* by unnecessarily imposing shared $w_{i,c}$ for all classes $y_c$, which fundamentally limits model expressivity.

**Proposition 2.** *Let $\mathcal{X}$ be the image space and $\mathcal{Y}$ be the class space. Given prompt set $\mathbb{P}$, for any prompt reweighting scheme $S$, define the representable likelihood set $\mathcal{F}_S$ as:*

$$\mathcal{F}_S = \left\{ f : \mathcal{X} \times \mathcal{Y} \to \mathbb{R}_+ \,\Big|\, \exists\, \mathbf{W} \in \mathcal{W}_S, \; \mathbb{P}, \;\; s.t. \quad f(\boldsymbol{x}, y_c) \;\propto\; \Pr(\boldsymbol{x} \mid y_c, \mathbf{W}, \mathbb{P}) \right\},$$

*where $\mathcal{W}_S$ is the weight space under scheme $S$. Let $\mathcal{F}_{CI}$ and $\mathcal{F}_{CS}$ be the representable likelihood sets induced from class-independent weighting (i.e., WPE) and class-specific weighting (cf. Eq. 1) schemes, respectively. Then, we have: $\exists f^* \in \mathcal{F}_{CS}$ such that $\forall f_{\mathrm{CI}} \in \mathcal{F}_{CI}, \exists \boldsymbol{x} \in \mathcal{X}, y_c \in \mathcal{Y}$ where $f^*(\boldsymbol{x}, y_c) \neq f_{\mathrm{CI}}(\boldsymbol{x}, y_c)$. That is, $\mathcal{F}_{CI}$ is a strict subset of $\mathcal{F}_{CS}$.*

**Remark 1.** *Proposition 2 formally states that class-specific weighting allows for capturing a richer set of image-text relationships than class-agnostic ones. To maximize potential expressivity, prompt weights $w_{i,c}$ **must** be class-specific to ensure that each class benefits from the most relevant prompts.*

**Modeling Predictive Probability** $\Pr(y^*|\boldsymbol{x}^*, \mathbb{P}, \mathbb{D}, \mathbf{W})$**.** We now come to predicting the label $\hat{y}_*$ for the query image $\boldsymbol{x}_*$. As zero-shot classification is *training-free*, a practical way is to approximate full $\Pr(y^*|\boldsymbol{x}^*, \mathbb{P}, \mathbb{D}, \mathbf{W})$ with $\Pr(y^*|\boldsymbol{x}^*, \mathbb{P}, \widehat{\mathbf{W}})$, where $\widehat{\mathbf{W}}$ is a point estimate *derived from unlabeled data* $\mathbb{D}$, per our discussion in Eq. 5 and Eq. 7. By considering each prompt $p_i \in \mathbb{P}$, we have

$$\Pr(y^*|\boldsymbol{x}^*, \mathbb{P}, \widehat{\mathbf{W}}) = \sum_{p_i \in \mathbb{P}} \Pr(y^*|\boldsymbol{x}^*, p_i, \widehat{\mathbf{W}}) \propto \frac{\exp\left(\sum_{i=1}^{n} (w_{i,c}\,\boldsymbol{z}_{i,c}^{\mathrm{T}})^\top \cdot \boldsymbol{z}_*^{\mathrm{I}}\right)}{\sum_{c' \in 1, \ldots, C} \exp\left(\sum_{i=1}^{n} (w_{i,c'}\,\boldsymbol{z}_{i,c'}^{\mathrm{T}})^\top \cdot \boldsymbol{z}_*^{\mathrm{I}}\right)}. \tag{8}$$

By now, we have framed VLM-based zero-shot classification in a probabilistic framework (Eq. 2), justified class-aware prompt reweighting (Propositions 1 and 2), and interpreted how class prediction for a query image can be performed (Eq. 8) under this understanding.

## 4 CLASS-AWARE PROMPT REWEIGHTING FOR VLMS

Guided by the probabilistic principles from Sec. 3, we next introduce CARPRT, a minimalistic *training-free* method designed to compute class-specific weights for prompt ensembling in VLMs.

**Overview.** Given an unlabeled dataset $\mathbb{D} = \{\boldsymbol{x}_j\}_{j=1}^{m}$, an *unknown* class space $\mathcal{Y} = \{y_1, \ldots, y_C\}$, a *fixed* prompt set $\mathbb{P} = \{p_i\}_{i=1}^{n}$, and a pre-trained VLM accessed via forward inference only, CARPRT aims to find the optimal weight matrix $\mathbf{W}^* \in \mathbb{R}^{n \times C}$, where each column $\mathbf{W}_c^* = [w_{1,c}^*, \ldots, w_{n,c}^*]^\top$ denotes the relative importance of different prompts for a particular class $y_c$ and specifies the contribution of each prompt $p_i$ to the class representation, as with Problem 1. Recall that the key insight driving CARPRT is that optimal prompt weights should reflect the **semantic alignment** between prompts and class concepts. As depicted in Fig. 2, CARPRT implements this insight through two steps: *Score Calculation* and *Weight Calculation* (the algorithmic outline can be found in App. D).

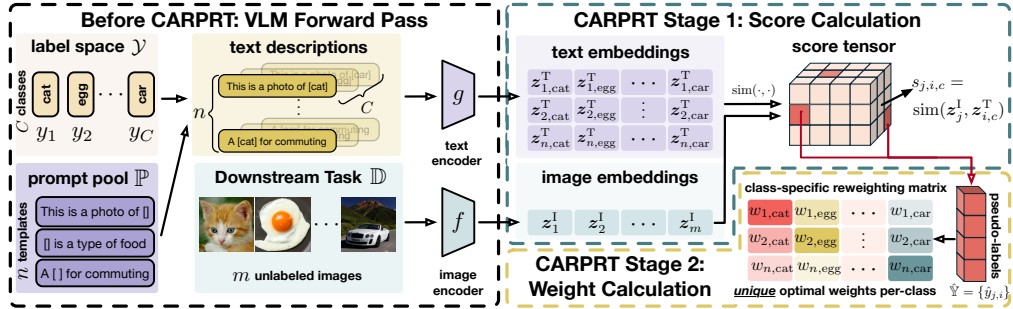

Figure 2: The CARPRT pipeline. First, the text encoder $g$ and image encoder $f$ yield textual class embeddings (from $C$ classes and $n$ prompts) and image embeddings (from $m$ unlabeled images). Then, compute the score tensor from image-text embedding similarities, each entry $s_{j,i,c}$ measures the relevance between the $i$-th prompt and the $j$-th image for the $c$-th class. Extract pseudo-labels from the score tensor, and derive the class-aware prompt reweighting matrix $\mathbf{W}$, which assigns class-specific weights for each prompt based on the scores.

**Stage 1: Prompt Relevance Score Calculation.** Eq. 3 and Eq. 4 suggest that estimating weight distribution $\Pr(\mathbf{W}|\mathbb{P}, \mathbb{D})$ hinges on the individual data likelihood $\Pr(\boldsymbol{x}_j|y_c, \mathbf{W}, \mathbb{P})$. As Lemma 1 established, $\Pr(\boldsymbol{x}_j|y_c, \mathbf{W}, \mathbb{P})$ is proportional to the VLM's similarity score, which is thus leveraged by CARPRT via forward similarity evaluations to compute raw similarity scores between all image embeddings and all prompt-derived text class embeddings. For an image $\boldsymbol{x}_j \in \mathbb{D}$, a prompt template $p_i \in \mathbb{P}$, and class $y_c \in \mathcal{Y}$, the relevance score $s_{j,i,c}$ is:

$$s_{j,i,c} = \text{sim}(\boldsymbol{z}_j^{\text{I}}, \boldsymbol{z}_{i,c}^{\text{T}}), \tag{9}$$

where $\boldsymbol{z}_j^{\text{I}} = f(\boldsymbol{x}_j)$ is the image embedding and $\boldsymbol{z}_{i,c}^{\text{T}} = g(p_i(y_c))$ is the text embedding for class $y_c$ under prompt $p_i$. This yields a *score tensor*, wherein each entry $s_{j,i,c}$ is an unnormalized estimate of $\Pr(\boldsymbol{x}_j|y_c, \mathbf{W}, \mathbb{P})$. The *score tensor* captures the semantic compatibility among all images $\mathbb{D}$, prompts $\mathbb{P}$, and classes $\mathcal{Y}$, providing the foundation for reweighting prompt-template combinations.

**Remark.** Eq. 9 is expressed using embeddings for convenience, but in practice we only query the VLM for the similarity scores $s_{j,i,c}$. Hence CARPRT works with a black-box CLIP interface and does not require explicit access to the embeddings.

**Stage 2: Class-Specific Weight Calculation.** The second stage transforms unnormalized similarity scores into normalized class-specific prompt weights through a process that *mirrors our probabilistic analysis* in Sec. 3. By empirically quantifying each prompt's relevance to specific classes, the resulting weights ensure that prompts primarily contribute to the aggregated representation of their most semantically aligned classes.

First, we create a pseudo-label set $\hat{\mathbb{Y}} = \{\hat{y}_{j,i}\}_{j=1,i=1}^{m,n}$ without any parameter update, by identifying, for each image-prompt pair, the class with the highest similarity score $\hat{y}_{j,i} = \arg\max_{y_c \in \mathcal{Y}} s_{j,i,c}$. Then, we calculate intermediate weight $w'_{i,c}$ for each prompt-class pair by aggregating the scores $s_{j,i,c}$ across all images $\boldsymbol{x}_j$ predicted to class $y_c$ under prompt $p_i$. This can be expressed as:

$$w'_{i,c} = \frac{\sum_{j=1}^{m} s_{j,i,c} \mathbb{1}_{\hat{y}_{j,i}=y_c}}{\sum_{j=1}^{m} \mathbb{1}_{\hat{y}_{j,i}=y_c}}. \tag{10}$$

Here, $\mathbb{1}_{\hat{y}_{j,i}=y_c}$ is the indicator function. Eq. 10 implements an empirical estimate of the class priors. $w'_{i,c}$ reflects the average strength of association prompt $p_i$ shows for class $y_c$ across $\mathbb{D}$, when $p_i$ itself identifies $y_c$ as the best match. Finally, these intermediate weights are normalized via

$$w^*_{i,c} = \frac{\exp\left(w'_{i,c}/\tau\right)}{\sum_{j=1}^{n} \exp\left(w'_{j,c}/\tau\right)}. \tag{11}$$

The temperature $\tau$ controls the sharpness of the distribution. This normalization ensures weights sum to one for each class, preserving their probabilistic validity. By constructing $w^*_{i,c}$ in this way, we integrate empirical class distributions into the reweighting scheme, ensuring that $w^*_{i,c}$ reflects

Table 1: Accuracy (%) comparison between baselines and our method % on various fine-grained classification datasets using CLIP and DeCLIP backbones. **Bold** values indicate the highest accuracy, while underlined values represent the second highest in each column. [*] "Human Selection" uses handcrafted prompts recommended by CLIP authors and introduces external knowledge. Results are not directly comparable to automated methods.

| | Caltech101 | DTD | EuroSAT | Aircraft | Food101 | Flower102 | Pets | Cars | SUN397 | UCF101 | ImageNet | Average |
|---|---|---|---|---|---|---|---|---|---|---|---|---|
| | | | | | CLIP-ViT-B/16 | | | | | | | |
| MPE | 92.50 | 46.88 | 51.86 | 21.49 | 85.34 | 64.21 | 79.46 | 65.21 | 64.92 | 67.41 | 67.59 | 64.26 |
| Majority Vote | 93.10 | 46.75 | 52.07 | 22.93 | 85.60 | 67.20 | 81.27 | 64.93 | 65.75 | 68.30 | 67.98 | 65.08 |
| WPE | 93.09 | 47.04 | 49.60 | 23.28 | 86.14 | 66.60 | 82.38 | 65.93 | 65.77 | 68.33 | 68.28 | 65.13 |
| CARPRT (Ours) | **94.16** | **48.90** | **55.56** | **24.49** | **86.31** | **71.36** | **89.13** | **66.14** | **66.93** | **70.41** | **68.59** | **67.45** |
| Human Selection[*] | 92.94 | 44.39 | 47.60 | 24.72 | 86.06 | 71.23 | 88.91 | 65.32 | 62.50 | 66.75 | 68.31 | 65.34 |
| | | | | | CLIP-ResNet50 | | | | | | | |
| MPE | 86.41 | 41.69 | 30.34 | 16.05 | 75.53 | 56.95 | 75.98 | 55.74 | 59.32 | 60.06 | 59.12 | 56.11 |
| Majority Vote | 86.79 | **42.14** | 28.86 | 16.29 | 76.00 | 60.06 | 77.29 | 56.01 | 60.40 | 60.87 | 59.24 | 56.72 |
| WPE | 86.65 | 40.89 | 30.65 | 16.11 | 76.15 | 58.82 | 78.43 | 56.02 | 59.71 | 61.53 | 59.78 | 56.79 |
| CARPRT (Ours) | **88.46** | 41.31 | **36.84** | **16.88** | **76.88** | **65.56** | **85.69** | **56.44** | **61.28** | **63.66** | **59.98** | **59.36** |
| Human Selection[*] | 86.29 | 40.32 | 29.56 | 17.28 | 75.31 | 66.14 | 85.77 | 55.61 | 58.52 | 61.46 | 59.71 | 57.82 |
| | | | | | DeCLIP-ViT-B/32 | | | | | | | |
| MPE | 94.04 | 41.63 | 28.05 | 7.10 | 71.71 | 77.76 | 76.75 | 52.22 | 62.08 | 57.87 | 67.01 | 57.84 |
| Majority Vote | 94.26 | 40.29 | 27.68 | 7.70 | 72.34 | 78.19 | 77.75 | 51.87 | 62.86 | 58.20 | 67.24 | 58.03 |
| WPE | 94.08 | 40.97 | 27.92 | 7.54 | 73.15 | 81.32 | 80.92 | 52.21 | 63.23 | 58.91 | 67.97 | 58.93 |
| CARPRT (Ours) | **94.37** | **43.31** | **33.14** | **8.76** | **74.15** | **82.42** | **83.28** | **52.23** | **64.12** | **59.57** | **68.08** | **60.31** |
| Human Selection[*] | 93.97 | 42.55 | 30.07 | 9.05 | 73.59 | 83.41 | 83.14 | 50.77 | 63.14 | 58.70 | 67.85 | 59.66 |

both the relevance scores (Eq. 4) and the estimated class priors (Eq. 5), thus providing a principled inference time approach to achieve *class-aware prompt reweighting*.

**Inference with Estimated Weights.** After calculating class-specific prompt weights $w_{i,c}^*$, we perform zero-shot inference by aggregating prompt relevance scores for each class. Given a test image $x^*$ with embedding $z_*^I = f(x^*)$ and prompt text embeddings $z_{i,c}^T = g(p_i(y_c))$, we first compute $s_{*,i,c} = \text{sim}(z_*^I, z_{i,c}^T)$ following Eq. 9. We then define the class score and prediction as

$$\hat{y}(x^*) = \arg \max_{c \in \{1,\dots,C\}} s_c(x^*), \qquad s_c(x^*) = \sum_{i=1}^{n} w_{i,c}^* \, s_{*,i,c}. \qquad (12)$$

This inference step only requires the similarity scores returned by the VLM (score-only black-box queries) and does *not* require access to model parameters, gradients, or internal embeddings.

**(Optional): Iterative Refinement.** While the single-pass pipeline described above forms the core of our approach, CARPRT can naturally be extended to refine both pseudo-labels and weights, by following the procedure *iteratively*: (i). Use current weight estimates to combine predictions from all prompts into refined pseudo-labels; (ii). Update class-specific weights based on these refined pseudo-labels. Importantly, this refinement procedure is *gradient-free* and thus does *not* require access to ground-truth labels. This alternating refinement process allows CARPRT to sharpen its weight estimates as pseudo-label quality improves. Full details are in App. E.1.

## 5 EXPERIMENTS

We evaluate how CARPRT performs on *zero-shot classification* with ten fine-grained benchmarks, compared to existing *prompt ensembling* methods. Our investigation centers on three questions: **(RQ1)** Does class-aware prompt reweighting outperform class-agnostic ones; if so, does it generalize across different VLM architectures and backbones? **(RQ2)** What factors contribute to CARPRT's effectiveness? **(RQ3)** Can CARPRT's benefit extend beyond zero-shot classification?

### 5.1 EXPERIMENTAL SETUP

**Dataset**. We evaluate on eleven classification benchmarks spanning diverse visual domains: Caltech101, DTD, EuroSAT, Aircraft, Food101, Flowers102, Pets, Cars, Sun397, UCF101 and ImageNet (details in App. C.1). We follow the evaluation protocol established by Zhou et al. (2022b).

**Models and Prompts**. We test CARPRT with three configurations: CLIP (Radford et al., 2021) with ViT-B/16 and ResNet50 backbones, and DeCLIP (Li et al., 2022) with the ViT-B/32, to validate if CARPRT generalizes across both CNN-based (He et al., 2016) and transformer-based (Dosovitskiy et al., 2021) backbones, and different VLM architectures. For all experiments, we use the same fixed set of 247 prompt templates from Allingham et al. (2023) to ensure fair comparisons.

**Baselines.** We compare CARPRT against three automated PE baselines: (1) MPE (Radford et al., 2021): Uniformly averages embeddings from all prompts. (2) Majority Vote (Allingham et al., 2023): Final prediction is based on the most frequent class predicted by individual prompts. (3) WPE (Allingham et al., 2023): Estimates a class-agnostic set of prompt weights from unlabeled test data. As an upper-bound reference, we also report "Human Selection" which uses a subset of prompts *manually filtered* for each dataset by human experts. This helps to benchmark automated methods against careful prompt engineering. See App. C.2 for details.

**Implementation**. We follow the publicly available code of baselines, with two adjustments noted. We use a smaller batch size for weight estimation due to resource limitations, and we omit its original frequency normalization step, which requires the external LAION-400M dataset (Schuhmann et al., 2021), since this step is not the focus of this study (See App. G.6 for the analysis of the impact). Moreover, this omission ensures all methods align with our problem setting of using *only unlabeled test data* without external resources, for fair comparison. Details and code are in App. C.3.

## 5.2 RESULTS OF ZERO-SHOT CLASSIFICATION

**Overall Comparison**. Tab. 1 shows that CARPRT consistently achieves the best accuracy across both fine-grained benchmarks and large-scale real-world datasets, such as ImageNet (with further evaluations on its variants provided in App. G.3). Gains are pronounced on datasets like Flower102 and Pets, highlighting the substantial impact of class-specific prompt relevance. Notably, CARPRT also surpasses Human Selection, where task-relevant prompts are manually filtered. This confirms that capturing class-specific weights can effectively *compensate for irrelevant prompts in generic prompt pools* and potentially outperform dataset-specific manual prompt engineering.

**Generalization and Robustness.**

*Across architectures.* CARPRT's performance benefits are consistent across different VLM architectures and backbones. With CLIP-ResNet50, despite its lower capacity than ViT-B/16, CARPRT still achieves clear and measurable gains. When applied to DeCLIP-ViT-B/32, which adopts a distinct pre-training strategy, CARPRT likewise maintains its strong lead. Overall, performance across diverse model configurations suggests that CARPRT can effectively *capture semantic relationships*, rather than exploiting a particular setup. Moreover, because it operates in a training-free, black-box manner, CARPRT is readily applicable whenever a VLM exposes only forward similarity queries.

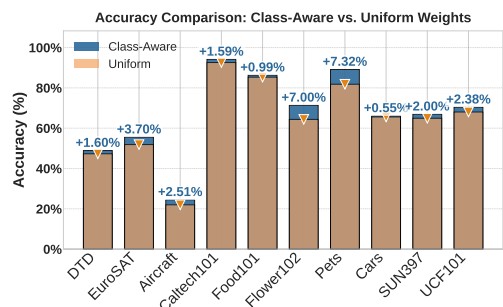

Figure 3: Accuracy gains of CARPRT over CARPRT-Uniform.

*Under distribution shifts.* We further evaluate robustness under distribution shifts on ImageNet and four variants: ImageNet-A, ImageNet-R, ImageNet-Sketch, and ImageNet-V2. In this setting, prompt weights are estimated once using only unlabeled samples from the in-distribution ImageNet test set, and the same weights are directly transferred to all variants for evaluation, without access to their target distributions during estimation. As shown in Tab. 2, CARPRT consistently surpasses MPE and WPE across all variants. These results suggest that CARPRT's reweighting strategy generalizes well under distribution shifts. We hypothesize that this robustness stems from estimating prompt–class relevance primarily via image–text score statistics tied to class semantics, and that ImageNet's larger sample size yields more stable weight estimates that transfer effectively across distributions.

Overall, these results indicate that CARPRT generalizes across both model architectures and data distributions, making it a practical plug-in for zero-shot classification in training-free, score-only

Table 2: Accuracy (%) under distribution shifts on ImageNet and its variants using CLIP ViT-B/16. Prompt weights are estimated once on in-distribution ImageNet and directly transferred to ImageNet-A, ImageNet-R, ImageNet-Sketch, and ImageNet-V2. **Bold** values indicate the highest accuracy in each column.

| Method | ImageNet | ImageNet-A | ImageNet-R | ImageNet-Sketch | ImageNet-V2 | Average |
|---|---|---|---|---|---|---|
| MPE | 67.59 | 49.35 | 77.33 | 46.92 | 61.37 | 60.51 |
| Majority Vote | 67.98 | 49.47 | 77.54 | 47.18 | 61.55 | 60.74 |
| WPE | 68.28 | 50.34 | 77.34 | 47.50 | 61.96 | 61.08 |
| CARPRT (Ours) | **68.59** | **51.96** | **77.69** | **47.91** | **62.51** | **61.73** |

black-box settings, where only forward similarity queries are available and no access to model parameters, gradients, or internal embeddings is required.

**Dataset-Specific Patterns**. The extent of CARPRT's improvement varies by dataset, showing larger gains on datasets with well-separated semantic categories (e.g., Flowers102, Pets). On highly specialized domains like Aircraft, the gains are modest, likely due to (i) the quality of the initial pseudo-labels generated by base VLMs, which impact both WPE and CARPRT. (ii) the suitability of generic prompt pool for highly specialized visual distinctions. Nonetheless, CARPRT consistently improves performance, highlighting the broad value of class-specific weighting.

## 5.3 ABLATION STUDY AND HYPERPARAMETER ANALYSIS

**Role of Class-specific Weights**. To isolate the benefit of class-specificity, we compare CARPRT to "CARPRT-Uniform". This variant first computes CARPRT's class-specific weights, then averages them across classes to yield a global $w_i^{\mathrm{u}} = \frac{1}{C} \sum_c w_{i,c}$ for each prompt $p_i$. This variant retains CARPRT's prompt scoring mechanism but discards class-level adaptation (it still differs from WPE; see App. G.2). As Fig. 3 shows, CARPRT consistently outperforms CARPRT-Uniform, with an average gain of 2.39%. Considerable improvements on datasets like Pets and Flowers102 affirm that tailoring prompt weights to individual classes is key to performance.

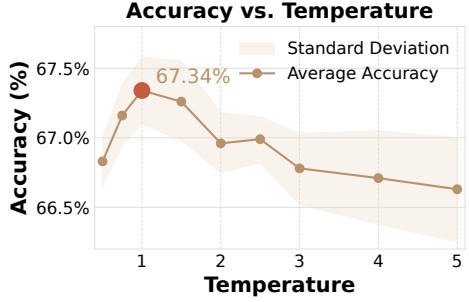

Figure 4: The variation of inference accuracy as the temperature $\tau$ changes, using CLIP-ViT-B/16.

**Temperature Sensitivity**. CARPRT uses a temperature $\tau$ (Eq. 11) to adjust prompt weight distributions. As shown in Fig. 4, $\tau = 1.0$ balances relevance and diversity, emphasizing useful prompts while preserving ensemble variety for generalization. Lower $\tau < 1.0$ concentrates weights on dominant prompts but reduces diversity, whereas higher values flatten the distribution. Although finding a single best hyperparameter for all zero-shot tasks is difficult, $\tau = 1.0$ is a stable choice across tasks, showing that calibrated reweighting helps without extensive per-task tuning. See App. G.2 for details.

## 5.4 EXTENDED EVALUATIONS AND CLASS-SPECIFIC WEIGHT VISUALIZATIONS

We explore CARPRT's versatility further with additional experiments (detailed in App. E,F,G).

**Refined Pseudo-Labels and Weight Estimation**. CARPRT's gains vary by dataset, partly due to the quality of initial pseudo-labels from the base VLM. Motivated by this observation, we further examine filtering low-confidence pseudo-labels (confidence-/entropy-based) to improve pseudo-label quality, but observe only marginal and inconsistent gains, suggesting that explicit filtering is unnecessary in practice (App. G.1). Instead, iterative refinement yields steady improvements by progressively leveraging increasingly accurate class information (App. E.2).

**Does Prompt Quality Matter?** While CARPRT is designed for *generic* prompt pools, it could further benefit from higher-quality, potentially domain-specific prompt templates. Preliminary tests with LLM-generated prompts showed improved CARPRT performance compared to using only dataset-agnostic templates from Allingham et al. (2023) (App. G.5), suggesting that CARPRT effectively

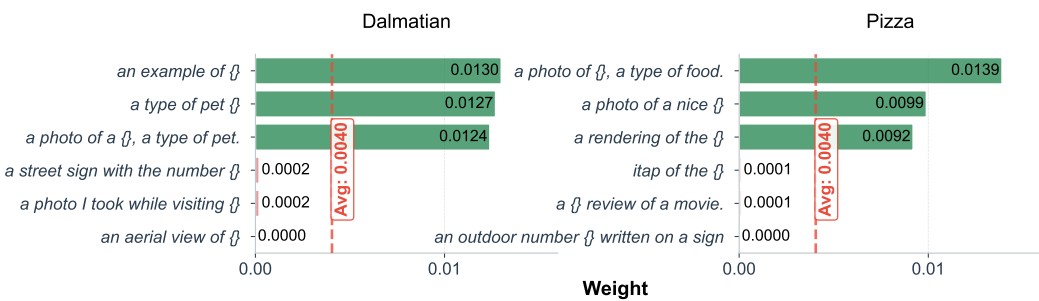

Figure 5: Visualization of class-specific prompt weights on Caltech101. For dalmatian and pizza, CARPRT assigns high weights to class-relevant prompts while suppressing irrelevant ones.

leverages the information in *any* given prompt set. While it is difficult to evaluate the "prompt quality", we argue that investing in careful prompt engineering is likely to be beneficial.

**CARPRT beyond Zero-shot Classification as a General-Purpose Plug-In**. We lastly show CARPRT's versatility as a component to enhance various VLM adaptation settings: (i) with *test-time adaptation* (Karmanov et al., 2024), CARPRT offers improved weight initialization (App. F.1); (ii) with *image-feature focused zero-shot methods* (Qian et al., 2024a), CARPRT enhances pseudo-labels for visual proxy learning (App. F.3); (iii) with *soft prompt tuning* (Lu et al., 2022), class-aware reweighting of learned prompts can boost performance further (App. F.2); (iv) with *LLM-empowered prompt augmentation* (Shtedritski et al., 2023; Mirza et al., 2024), the utility of high-quality generated prompts can still be improved via class-aware reweighting (App. F.4). All these results confirm CARPRT's flexibility as a general-purpose plug-in for broader VLM adaptation scenarios.

**Visualization of Class-Specific Prompt Weights.** To provide qualitative insight into CARPRT's mechanism, we visualize class-specific prompt weights on Caltech101. Fig. 5 shows weights estimated for two representative classes, *dalmatian* and *pizza*. For *dalmatian*, CARPRT assigns higher weights to semantically relevant prompts (e.g., example, pet, photo) while suppressing mismatched ones (e.g., aerial, visiting, number). Similarly, for *pizza*, food-related prompts (e.g., food, photo, rendering) are prioritized, whereas unrelated terms (e.g., sign, movie, itap) are down-weighted. Overall, these visualizations corroborate our quantitative results and illustrate that CARPRT learns class-dependent prompt preferences; additional examples are provided in App. J.

## 6  DISCUSSION AND FUTURE OUTLOOK

**Broader Related Works**. The performance of VLM adaptation in downstream classification tasks is relevant to the text prompt, motivating research on improving prompt effectiveness in *different directions*. *Prompt tuning* (Zhou et al., 2022b; Khattak et al., 2023a) optimizes task-specific soft prompts through training, but departing from zero-shot settings. *Unsupervised transfer learning* methods (Qian et al., 2024a) aim to bridge domain gaps between visual and textual embeddings without labels; they do not focus on combining multiple prompts. *Augmentation-based weighting* instead relies on large-scale data augmentation, such as using LLMs to generate task-specific prompts or building partial image views, then assigning weights to augmented prompts or views (Zhu et al., 2024; Li et al., 2024); while powerful, they necessitate the availability of external computing resources. In contrast, CARPRT explicitly addresses the setting of *prompt ensembling* with a fixed, potentially task-irrelevant prompt pool. It is entirely *training-free*, relies on neither label supervision nor LLM-generated prompts, and focuses on reweighting existing prompts to capture class-specific relevance. This makes CARPRT *orthogonal* to the above directions, while also complementary to them, offering a unique perspective on VLM adaptation. We discuss these related works in detail in App. A.

**Summary**. This study focused on prompt ensembling and confirmed that class-aware prompt reweighting is not only beneficial but essential for improving the efficacy of VLMs across a variety of downstream classification tasks. By moving beyond uniform weighting, we showed that adapting weights to better reflect the class-specific characteristics leads to measurable gains in performance. We hope this study encourages further exploration of integrating class-awareness with other VLM adaptation techniques to enhance across a wider range of applications.

## ETHICS STATEMENT

All authors have read and agree to abide by the ICLR Code of Ethics and Code of Conduct. This work does not involve sensitive personal data or experiments with human subjects. We have taken care to ensure that the datasets used are publicly available and widely adopted in prior research, and that the proposed method does not raise foreseeable ethical concerns. All claims and findings are reported honestly and transparently.

## REPRODUCIBILITY STATEMENT

We are committed to ensuring the reproducibility of our results. Detailed descriptions of the setup, training and evaluation protocols, implementation details, and hyperparameter settings are provided in Sec. 5 and App. C.3. All experiments are conducted on publicly available datasets, which are listed in App. C.1. An anonymous code link is supplied to facilitate replication.

## AUTHOR CONTRIBUTIONS

All authors contributed to the research and the writing of this paper.

## ACKNOWLEDGMENTS

We thank the anonymous reviewers and area chairs for their constructive feedback. MS was supported by JST ASPIRE Grant Number JPMJAP25B1. Jianzhong Qi is supported in part by the Australian Research Council (ARC) via Discovery Project DP240101006 and Future Fellowship FT240100170. Feng Liu is supported by the Australian Research Council (ARC) with the grant number DE240101089, LP240100101, DP230101540 and the NSF&CSIRO Responsible AI program with grant number 2303037.

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

## A    DETAILED DISCUSSION ON RELATED WORKS

**Prompt Tuning Methods.**    Prompt tuning adapts a pre-trained model by introducing learnable embeddings (prompt tokens) at the input stage. These tokens can be instantiated as textual prompts or visual prompts, enabling task-specific adaptation through the model's input interface. CoOp first applied prompt tuning to CLIP by optimizing learnable prompts in the text branch for few-shot recognition (Zhou et al., 2022b). To address CoOp's limited generalization, CoCoOp conditionally generates prompts from visual features (Zhou et al., 2022a). MaPLe further extends prompt tuning to both the vision and text branches to improve transferability (Khattak et al., 2023a). Building on MaPLe, PromptSRC enhances prompt learning by leveraging descriptive text generated by large language models (LLMs), e.g., GPT-4 (Khattak et al., 2023b). However, these tuning-based approaches require optimizing learnable variables (and typically labeled downstream data, even in the few-shot regime), which falls outside our strict zero-shot, training-free setting. We therefore do not include them as baselines for CARPRT.

**Unsupervised Transfer Methods for VLMs.**    Unsupervised transfer for VLMs aims to adapt pre-trained models (e.g., CLIP) to downstream tasks without ground-truth labels. Existing methods can be broadly grouped into two directions. The first direction, exemplified by Weighted Prompt Ensembling (WPE) (Allingham et al., 2023), focuses on *automatically reweighting prompts from a given template pool*. By assigning dataset-level importance weights to different templates, such methods help identify which prompts are more compatible with a downstream task and can offer a degree of interpretability (Allingham et al., 2023). The second direction relies on transductive learning using image features to construct classifiers, such as InMaP (Qian et al., 2024a). These methods often achieve stronger accuracy by exploiting structure in the unlabeled images, but are typically less interpretable since decisions are driven primarily by image-feature learning rather than explicit prompt contributions. Our work follows the first direction by retaining the prompt-based formulation while improving accuracy via class-specific reweighting. Moreover, the pseudo-labels produced by CARPRT can also benefit image-centric transductive pipelines; see App. F.3 for detailed results.

**View-aware Weighting Approaches.**    View-aware methods improve zero-shot transfer by aggregating evidence from multiple augmented visual or textual views and weighting them by confidence or alignment. WCA performs local visual prompting by pooling similarities between cropped regions and fine-grained textual descriptions using weighted aggregation (Li et al., 2024). AWT combines diverse image augmentations with LLM-generated prompts and computes weights across views, followed by optimal transport for cross-modal alignment (Zhu et al., 2024). While effective, these approaches may rely on external resources (e.g., LLMs) and/or costly augmentations at inference. In contrast, CARPRT derives class-specific weights directly from image–text similarity scores under a fixed prompt pool, without external models or additional augmentations, yielding substantially lower inference overhead while remaining complementary to view-aware techniques.

**Test-time Adaptation.**    Test-time adaptation (TTA) aims to adapt models to unlabeled test data (Ganin et al., 2016; Long et al., 2015; Zhang et al., 2022). Broadly, TTA methods can be categorized into training-based and training-free approaches. Training-based methods update model parameters or prompts using test-time objectives (e.g., entropy minimization), as in TENT (Wang et al., 2021), or incorporate additional regularization to preserve alignment, as in CoTTA (Chen et al., 2022). For VLMs, prompt-centric TTA methods such as TPT fine-tune a learnable prompt at test time (Shu et al., 2022), and DiffTPT further leverages diffusion models to enrich test-time augmentations (Feng et al., 2023). Training-free TTA methods instead rely on adjusting normalization statistics or test-time augmentation without updating model parameters (Li et al., 2016; Karmanov et al., 2024). Since CARPRT is also training-free and uses only unlabeled test data, we study its relationship with training-free TTA in App. F.1.

## B    DIFFERENT PROBLEM SETUP FOR VLMS ADAPTATION

Prompt ensembling, as formalized in Problem 1, targets a strictly zero-shot inference setting where the only available resources are a fixed prompt template set $\mathbb{P}$ and an unlabeled test set $\mathbb{D}$. No

learnable parameters, task-specific fine-tuning, or external supervision are permitted. This setting is entirely inference-time, model-free, and tuning-free.

In contrast, other VLM adaptation paradigms operate under more relaxed assumptions, either by enabling trainable components, leveraging supervision, or utilizing additional knowledge sources. We outline the key differences as follows:

**Prompt Tuning** relaxes the "no training" constraint by introducing learnable prompt tokens, typically optimized using downstream supervision. Formally, the prompt becomes a learnable function $p_\theta(y_c)$ with parameters $\theta$, where $\theta$ is optimized on labeled data $\{(x_j, y_j)\}$. CoOp Zhou et al. (2022b) learns a global soft prompt, while CoCoOp Zhou et al. (2022a) further conditions it on image embeddings $f(x)$ to improve generalization. These methods trade interpretability for adaptability and require supervision at training time.

**LLM-Generated Descriptions** expand the prompt space $\mathcal{P}$ using external generative models. Rather than fixing $\mathcal{P}$ a priori, a large language model $g_{\text{LLM}}$ generates class descriptions $\tilde{p}_i(y_c) = g_{\text{LLM}}(y_c)$ that are often more expressive and context-aware Menon & Vondrick (2023a). While such prompts can improve alignment, this introduces non-negligible computational overhead and reduces reproducibility, especially when prompts are generated on-the-fly.

**Image-Centric Adaptation** bypasses prompt usage entirely by constructing classifiers purely from image features. Methods like InMaP Qian et al. (2024b) rely on clustering method to construct a label assignment function $h : \mathcal{X} \to \mathcal{Y}$ without accessing any textual information. These methods often outperform prompt-based approaches in raw accuracy but offer limited interpretability and are incompatible with text-conditioned decision-making.

CARPRT operates strictly within the constraints of Problem 1. Unlike the above paradigms, it does not rely on any learnable components, LLM-generated text, or image-only inference. Instead, it focuses on exploiting the class-specific alignment between $\mathcal{P}$ and $\mathcal{Y}$ in a training-free, interpretable, and modular fashion. As demonstrated in App. F, its output (pseudo-labels and weights) can directly benefit and enhance downstream methods in both prompt tuning and image-centric learning pipelines.

## C  DATASETS, BASELINE METHODS, AND IMPLEMENTATION

### C.1  DATASETS

**Fine-grained Datasets.**  Following Zhou et al. (2022b), we evaluate our method on 10 fine-grained classification benchmarks: Caltech101 (Fei-Fei et al., 2004) (101 object categories); DTD (Cimpoi et al., 2014) (47 describable texture attributes such as "bumpy" and "scaly"); EuroSAT (Helber et al., 2019) (10 land-use classes from satellite imagery, e.g., residential, forest, and river); FGVC-Aircraft (Maji et al., 2013) (100 aircraft variants); Food101 (Bossard et al., 2014) (101 food categories); Flowers102 (Nilsback & Zisserman, 2008) (102 flower species); Oxford Pets (Parkhi et al., 2012) (37 pet breeds); Cars196 (Krause et al., 2013) (196 car models); SUN397 (Xiao et al., 2010) (397 scene categories); and UCF101 (Khurram, 2012) (101 human action classes).

**ImageNet and its Variants.**  Following Allingham et al. (2023), we additionally evaluate on ImageNet and its commonly used robustness variants: ImageNet (Russakovsky et al., 2015) (1,000 classes); Tiny-ImageNet (Le & Yang, 2015) (a 200-class subset); ImageNet-A (Hendrycks et al., 2021b) (naturally adversarial examples); ImageNet-R (Hendrycks et al., 2021a) (renditions such as paintings and cartoons); ImageNet-Sketch (Wang et al., 2019) (sketch-style images); and ImageNet-V2 (Recht et al., 2019) (a re-collected test set).

### C.2  BASELINES

To evaluate our method under a consistent setting, we compare CARPRT with representative baselines that operate within the same zero-shot classification protocol and fixed prompt set (Problem 1).

**Mean Prompt Ensembling (MPE).** MPE averages predictions across prompts with uniform weights. For each class, the model encodes all prompted class texts and averages them to form a class prototype.

---

**Algorithm 1** Class-Aware Prompt Reweighting (CARPRT)

---

**Input:** A pre-trained VLM model $\Phi(\boldsymbol{x}, p_i(y_c))$ that returns a cosine similarity score[3], a prompt set $\mathbb{P}$, an unlabeled dataset $\mathbb{D}$, a candidate label space $\mathcal{Y}$, the temperature parameter $\tau$ and the normalization scale $\lambda$.

**1: Construct** prompted-class texts $p_i(y_c), \forall p_i \in \mathbb{P}, \forall y_c \in \mathcal{Y}$;

**2: Obtain** the relevance score set $\mathbb{S} = \{s_{j,i,c}\}_{j=1,i=1,c=1}^{m,n,C}$, by querying the scorer: $s_{j,i,c} = \Phi(\boldsymbol{x}_j, p_i(y_c))$;

**3: Obtain** the pseudo-label set: $\hat{\mathbb{Y}} = \{\hat{y}_{j,i}\}_{j=1,i=1}^{m,n}$;

**4: Derive** the weight matrix $\mathbf{W}^*$ by Eq. 10 and Eq. 11;

**Output:** a class-aware prompt weight matrix $\mathbf{W}^*$.

---

At test time, an image is classified by cosine similarity to these averaged embeddings. This baseline assumes prompts contribute equally, regardless of class or semantics.

**Majority Vote.** Majority Vote treats each prompt as an independent voter. For each prompt, the model predicts the most similar class for an image, and the final prediction is determined by majority voting across prompts. This method ignores prediction confidence and assumes all prompt votes carry equal importance.

**Weighted Prompt Ensembling (WPE)** (Allingham et al., 2023). WPE estimates a global prompt-weight vector from the unlabeled test set and aggregates prompt-conditioned class embeddings accordingly (e.g., via an unsupervised objective such as entropy minimization). However, WPE uses a single weight vector shared across all classes and thus cannot capture class-specific variations in prompt relevance.

## C.3 DETAILS REGARDING EXPERIMENTS

**Implementation Details.** We implement all methods using PyTorch 1.7.1 and Python 3.7.6, and run experiments on a single NVIDIA A100 Tensor Core GPU. We use OpenAI CLIP (Radford et al., 2021) and DeCLIP (Li et al., 2022) as the underlying VLM backbones. Our code is available at `https://github.com/tmlr-group/CARPRT` for reproducibility.

**Hyperparameter Settings.** Unless otherwise specified, we set $\tau = 1.0$ for fine-grained datasets and $\tau = 1.5$ for ImageNet (Russakovsky et al., 2015) and its variants, and use a batch size of 512 for all experiments.

## D MORE DETAILS OF CARPRT

### D.1 CARPRT ALGORITHM

We summarize the overall procedure of our proposed Class-Aware Prompt Reweighting (CARPRT) in Algorithm 1. As shown in the algorithm, CARPRT begins by encoding both image and text embeddings using a pre-trained CLIP-liked model. It then computes the relevance score between image features and prompt-conditioned text features, followed by pseudo-label assignment. Finally, a class-aware weight matrix is derived based on the computed scores, enabling the construction of a refined prompt weight matrix that improves zero-shot classification performance.

### D.2 CONNECTING CARPRT FORMULATION WITH THE PROBABILISTIC FRAMEWORK

We now detail the correspondence between the CARPRT formulation (Section 4) and the probabilistic framework established in Section 3.

---

[3]We write CARPRT using a black-box scoring function $\Phi$ since the method only requires access to similarity scores (or logits) $s_{j,i,c}$ from a forward pass. Explicit access to the image/text encoders (and embeddings) is only needed if one wishes to optimize alternative objectives beyond such score queries (e.g., non-cosine similarity losses or representation-level regularizers).

---

**Algorithm 2** Iterative Class-Aware Prompt Reweighting (iCARPRT)

---

**Input:** A pre-trained VLM model $\Phi(\boldsymbol{x}, p_i(y_c))$ that returns a cosine similarity score, a prompt set $\mathbb{P}$, an unlabeled dataset $\mathbb{D}$, a candidate label space $\mathcal{Y}$, the maximum iterations $T_{max}$, the temperature parameter $\tau$ and the normalization scale $\lambda$.

**1: Generate** prompted-class texts $p_i(y_c), \forall p_i \in \mathbb{P}, \forall y_c \in \mathcal{Y}$;

**2: Obtain** the relevance score set $\mathbb{S} = \{s_{j,i,c}\}_{j=1,i=1,c=1}^{m,n,C}$ by querying the scorer: $s_{j,i,c} = \Phi(\boldsymbol{x}_j, p_i(y_c))$;

**3: Initialize** the class-aware weights $w_{i,c}^{(0)}$ uniformly;

**for** $t = 1$ *to* $T_{max}$ **do**

    **4: Obtain** the pseudo-labels set: $\hat{\mathbb{Y}} = \{\hat{y}_j\}_{j=1}^{m}$ using Eq. 13;

    **5: Derive** the weight matrix $\mathbf{W}^t$ by Eq. 14 and Eq. 11;

**end**

**Output:** a class-aware prompt weight matrix $\mathbf{W}^* = \mathbf{W}^{T_{\max}}$.

---

Concretely, the practical implementation Eq. 9– Eq. 11 align with Eq. 3– Eq. 7 in the following manner.

**Score Calculation**. Eq. 9 implements the likelihood term $\Pr(\boldsymbol{x}_j|y_c, W, \mathbb{P})$ from Eq. 7 by defining $s_{j,i,c} = \frac{\exp(a_{j,i,c}/\lambda)}{\sum_{y \in \mathcal{Y}} \exp(a_{j,i,c}/\lambda)}$. This formulation aligns with the EBM in Eq. 7 by using cosine similarity $a_{j,i,c}$ as the negative energy term and normalizing through softmax to obtain proper probabilities.

**Weight Calculation**. Eq. 10– Eq. 11 correspond to estimating $\Pr(W|\mathbb{P}, \mathbb{D})$ from Eq. 4 through a two-step process. Eq. 10 first obtains the pseudo-labels for samples as the empirical estimates $\widehat{\Pr}(y_c|W, \mathbb{P})$ (i.e., Eq. 5). It then estimates intermediate weights by aggregating scores across pseudo-labeled samples by multiplying the scores $\Pr(\boldsymbol{x}_j|y_c, W, \mathbb{P})$ (i.e., $s_{j,i,c}$) with $\widehat{\Pr}(y_c|W, \mathbb{P})$. Eq. 11 applies softmax to ensure the resulting weights form a valid probability distribution over prompts for each class, which satisfies the simplex constraint implied by our probabilistic framework.

# E   DETAILS OF CARPRT WITH ITERATIVE REFINEMENT (iCARPRT)

## E.1   METHODS

In this section, we introduce *iterative class-aware prompt reweighting* (iCARPRT). Unlike the single-pass approach described in the main text, iCARPRT refines pseudo-labels and class-aware prompt weights through multiple rounds of alternating updates. The procedure consists of the following two main steps: pseudo-label generation and class-aware weight estimation.

In pseudo-label generation, the pseudo-label $\hat{y}_j$ of the image $\boldsymbol{x}_j$ is computed using the prompt weights estimated in the previous iteration, $\mathbf{W}^{t-1}$, as:

$$\hat{y}_j = \arg\max_{y_c \in \mathcal{Y}} \sum_{i=1}^{n} w_{i,c}^{t-1} s_{j,i,c} \tag{13}$$

where $s_{j,i,c}$ is the relevance score computed in Eq. 9. Once the pseudo-labels $\hat{y}_j$ are updated, the intermediate weights $w_{i,c}'$ are estimated by:

$$w_{i,c}' = \frac{\sum_{j=1}^{m} s_{j,i,c} \mathbb{1}_{\hat{y}_j=y_c}}{\sum_j \mathbb{1}_{\hat{y}_j=y_c}}. \tag{14}$$

where $\mathbb{1}_{\hat{y}_j=y_c}$ is an indicator function that is 1 if $\hat{y}_j = y_c$, and 0 otherwise. Then the final weights $w_{i,c}^*$ are computed from the intermediate weights $w_{i,c}'$ using Eq. 11.

These two steps repeat until a predefined maximum number of iterations is reached. By alternating between pseudo-label prediction and weight re-estimation, iCARPRT creates a reinforcing cycle that continuously improves both the pseudo-labels and the class-aware prompt weights.

Table 3: Accuracy (%) comparison between CARPRT and iCARPRT on various fine-grained classification datasets using CLIP-ViT-B/16 and CLIP-ResNet50 backbones. **Bold** values indicate the highest accuracy.

| | Caltech101 | DTD | EuroSAT | Aircraft | Food101 | Flower102 | Pets | Cars | SUN397 | UCF101 | Average |
|---|---|---|---|---|---|---|---|---|---|---|---|
| | | | | | CLIP-ViT-B/16 | | | | | | |
| CARPRT | 94.16 | **48.90** | **55.56** | **24.49** | 86.31 | 71.36 | 89.13 | 66.14 | 66.93 | 70.41 | 67.34 |
| iCARPRT | **94.27** | 48.14 | 54.79 | 23.71 | **87.25** | **72.01** | **89.64** | **67.19** | **67.28** | **70.53** | **67.48** |
| | | | | | CLIP-ResNet50 | | | | | | |
| CARPRT | 88.46 | 41.31 | **36.84** | **16.88** | 76.88 | 65.56 | 85.69 | 56.44 | 61.28 | 63.66 | 59.30 |
| iCARPRT | **89.14** | **41.83** | 35.65 | 15.42 | **77.96** | **66.13** | **86.09** | **57.28** | **61.45** | **64.32** | **59.53** |

Table 4: Accuracy (%) comparison between baselines and CARPRT when combined with TDA, using CLIP-ViT-B/16 and CLIP-ResNet50 backbones. **Bold** values represent the highest accuracy in each column.

| | Caltech101 | DTD | EuroSAT | Aircraft | Food101 | Flower102 | Pets | Cars | SUN397 | UCF101 | Average |
|---|---|---|---|---|---|---|---|---|---|---|---|
| | | | | | CLIP-ViT-B/16 | | | | | | |
| MPE | 93.18 | 46.75 | 60.60 | 23.37 | 86.04 | 65.61 | 84.21 | 67.44 | 66.41 | 71.48 | 66.51 |
| WPE | 93.49 | 47.02 | 62.48 | 23.09 | 86.21 | 68.10 | 84.12 | 67.23 | 66.98 | 71.23 | 67.00 |
| CARPRT (Ours) | **94.62** | **48.52** | **63.95** | **24.05** | **86.50** | 70.36 | 84.50 | **67.83** | **68.06** | **71.85** | **68.02** |
| Human Selection (TDA) | 94.24 | 47.40 | 58.00 | 23.91 | 86.14 | **71.42** | 88.63 | 67.28 | 67.62 | 70.66 | 67.53 |
| | | | | | CLIP-ResNet50 | | | | | | |
| MPE | **92.03** | 41.77 | 54.56 | 19.77 | 83.41 | 62.50 | 80.65 | 63.55 | 64.14 | 68.80 | 63.12 |
| WPE | 91.67 | 41.89 | 56.78 | 19.84 | 83.21 | 56.67 | 81.66 | 64.87 | 64.87 | 68.72 | 63.45 |
| CARPRT (Ours) | 91.75 | **42.71** | **57.65** | 19.98 | **83.61** | 62.66 | 81.38 | **65.98** | **65.98** | **68.65** | **63.76** |
| Human Selection (TDA) | 91.42 | 41.00 | 56.97 | **20.55** | 83.34 | **62.75** | **83.62** | 64.14 | 65.86 | 68.52 | 63.82 |

## E.2 EXPERIMENTAL RESULTS

We evaluate the performance of iCARPRT against the single-pass version, CARPRT. As shown in Tab. 3, the results demonstrate that iCARPRT achieves improvements in mean accuracy across different backbones. This suggests that the iterative refinement process effectively enhances class-aware prompt weighting by progressively improving pseudo-label quality and weight estimation.

**Quality of Pseudo Labels Matters**. In datasets such as EuroSAT and Aircraft, iCARPRT does not outperform CARPRT. A possible reason is the relatively low initial pseudo-label accuracy in these datasets. Since iCARPRT updates prompt weights based on pseudo-labels in each iteration, a poor starting point may lead to reinforcement of incorrect labels rather than improvement. In such cases, the iterative updates fail to enhance pseudo-label quality, limiting the effectiveness of the approach.

## F COMBINING CARPRT WITH OTHER VISION-LANGUAGE METHODS

While CARPRT focuses on strict zero-shot image classification with a fixed set of handcrafted prompts and unlabeled data (Problem 1), it is inherently modular and can be integrated into a wide range of existing vision-language pipelines. Although direct comparison is not meaningful due to differing problem assumptions, we show that CARPRT can function as a complementary component rather than a competing method.

Specifically, we conduct case studies in three representative scenarios. We first combine CARPRT with a test-time adaptation method, then apply it to augment soft prompt tuning, and finally integrate it with a recent zero-shot method that leverages LLM-generated prompts. Details and results for each case are presented in the following subsections.

## F.1 COMBINING CARPRT WITH TEST-TIME ADAPTATION METHOD

CARPRT can be integrated with TDA, a state-of-the-art training-free test-time adaptation (TTA) method for CLIP that enables efficient adaptation without backpropagation (Karmanov et al., 2024).

Our approach is not in conflict with TDA but is orthogonal to it. While TDA uses a human-selected prompt pool for each task, our method can serve as a complementary module that replaces

this human selection pool, providing an alternative way of selecting prompts without requiring human intervention. This allows our method to work alongside TDA, enhancing the adaptability of vision-language models in a more automated manner. We conduct the experiment to compare the performance of our method with several baselines, including the human-selected prompts, the equal weight prompt selection, WPE, all combined with the TDA method. The results are evaluated using both CLIP-ViT-B/16 and CLIP-ResNet50 backbones across ten fine-grained datasets, as shown in Tab. 4.

From the result, we can observe that our method outperforms the other baselines in several datasets, achieving the highest average accuracy of 67.96% for CLIP-ViT-B/16 and 63.76% for CLIP-ResNet50. Specifically, for datasets like EuroSAT, Food101, and Flower102, our method shows significant improvements over the human-selected and WPE baselines. These improvements demonstrate that our approach effectively enhances the performance of TTA methods, by offering a more efficient prompt selection strategy. However, there are cases where it falls short compared to human-selected prompts. This may be caused by the limited diversity and smaller size of the template pool, where automatic reweighting methods may not perform as well as direct human selection. However, the automated approach significantly reduces the human labor cost. This experiment demonstrates the promising future of our method—not only in prompt reweighting but also as a technique that can be integrated into other vision-language model (VLM) transfer learning approaches. The ability to automatically adjust prompts in a computationally efficient manner paves the way for broader applications and adaptability in various VLM-based tasks.

**Posterior Update with TTA.** When prompt weights can be updated continuously, such as in TTA settings, different priors (e.g., uniform, global Dirichlet, or class-specific Dirichlet) define initial beliefs about weight distributions before observing test data. In the TTA scenario, test data arrives as a stream: $\{\boldsymbol{x}^{(0)}, \ldots, \boldsymbol{x}^{(t)}, \boldsymbol{x}^{(t+1)}, \ldots\}$. Based on Eq. 4, we have a general form of posterior

$$p(\mathbf{W}|\boldsymbol{x}^{(t)}, \mathbb{P}) \propto p(\boldsymbol{x}^{(t)}|\mathbf{W}, \mathbb{P})p(\mathbf{W}|\mathbb{P}),$$

where $p(W|\mathbb{P})$ is the prior, $p(\boldsymbol{x}^{(t)}|\mathbf{W}, \mathbb{P})$ is the likelihood from test data, and $p(W|\boldsymbol{x}^{(t)}, \mathbb{P})$ is the posterior that guides weight updates sample-by-sample. The posterior updating process follows:

For first test sample $\boldsymbol{x}^{(0)}$:

$$\text{Prior}: p(\mathbf{W}|\mathbb{P})$$
$$\text{Likelihood}: p(\boldsymbol{x}^{(0)}|\mathbf{W}, \mathbb{P})$$
$$\text{Posterior}: p(W|\boldsymbol{x}^{(0)}, \mathbb{P}) \propto p(\boldsymbol{x}^{(0)}|\mathbf{W}, \mathbb{P})p(\mathbf{W}|\mathbb{P})$$

Then, as we observe the second test sample $\boldsymbol{x}^{(1)}$, we have

$$\text{Prior}: p(\mathbf{W}|\boldsymbol{x}^{(0)}, \mathbb{P}) \text{ (previous posterior)}$$
$$\text{Likelihood}: p(\boldsymbol{x}^{(1)}|\mathbf{W}, \mathbb{P})$$
$$\text{Posterior}: p(\mathbf{W}|\boldsymbol{x}^{(0)}, \boldsymbol{x}^{(1)}, \mathbb{P}) \propto p(\boldsymbol{x}^{(1)}|\mathbf{W}, \mathbb{P})p(\mathbf{W}|\boldsymbol{x}^{(0)}, \mathbb{P})$$

This leads to the sequential update scheme, formulated as

$$p(\mathbf{W}|\boldsymbol{x}^{(0)}, \ldots, \boldsymbol{x}^{(t)}, \mathbb{P}) \propto p(\boldsymbol{x}^{(t)}|\mathbf{W}, \mathbb{P})p(\mathbf{W}|\boldsymbol{x}^{(0)}, ..., \boldsymbol{x}^{(t-1)}, \mathbb{P})$$

Thus, in TTA settings, these priors can be (1) initialized based on initial test samples; and (2) updated sequentially as new test samples arrive.

More specifically, choosing different prior distributions would lead to different updating computations.

*Uniform Prior.* Recall the uniform prior is defined as

$$p(W|\mathbb{P}) = \begin{cases} \frac{1}{|\mathcal{W}|} & \text{if } W \in \mathcal{W} \\ 0 & \text{otherwise} \end{cases}$$

By taking log to both LHS and RHS, we will have

$$\log p(\mathbf{W}|\mathbb{P}) = \begin{cases} -\log|\mathcal{W}| & \text{if } \mathbf{W} \in \mathcal{W} \\ -\infty & \text{otherwise} \end{cases}$$

which then leads to the log posterior to be expressed as

$$\log p(\mathbf{W}|\boldsymbol{x}^{(t)}, \mathbb{P}) \propto -\log|\mathcal{W}| + \log \sum_{y_c \in \mathcal{Y}} p(\boldsymbol{x}^{(t)}|y_c, \mathbf{W}, \mathbb{P})p(y_c|\mathbf{W}, \mathbb{P})$$

$$= -\log|\mathcal{W}| + \log \sum_{y_c \in \mathcal{Y}} \exp\left(\sum_{i=1}^{n} \left(w_{i,c}\boldsymbol{z}_{i,c}^{\mathrm{T}}\right)^{\top} \cdot \boldsymbol{z}^{\mathrm{I}}\right) \cdot \frac{\mathbb{1}_{\hat{y}_{ji}=y_c}}{\sum_{j'} \mathbb{1}_{\hat{y}_{j'i}=y_c}}$$

*Global Dirichlet Prior.* The global Dirichlet prior treats all weights across classes as a single vector:

$$p(W|\mathbb{P}) = \mathrm{Dir}(\mathrm{vec}(W)|\alpha_1, ..., \alpha_{nC})$$

where $\mathrm{vec}(\mathbf{W}) \in \mathbb{R}^{nC}$ is the vectorization of weight matrix W (here we denote $C = |\mathcal{Y}|$ as the cardinality of label space) Similarly, we will have the log prior and posterior as

$$\log p(\mathbf{W}|\mathbb{P}) = \log \mathrm{Dir}(\mathrm{vec}(\mathbf{W})|\alpha_1, ..., \alpha_{nC})$$

$$= \log \Gamma(\alpha_0) - \sum_{k=1}^{nC} \log \Gamma(\alpha_k) + \sum_{k=1}^{nC} (\alpha_k - 1) \log w_k \quad (\alpha_0 = \sum_{k=1}^{nC} \alpha_k)$$

$$= \log \Gamma(\sum_{k=1}^{nC} \alpha_k) - \sum_{c=1}^{C} \sum_{i=1}^{n} \log \Gamma(\alpha_{(c-1)n+i}) + \sum_{c=1}^{C} \sum_{i=1}^{n} (\alpha_{(c-1)n+i} - 1) \log w_{i,c}$$

and

$$\log p(\mathbf{W}|\boldsymbol{x}^{(t)}, \mathbb{P}) \propto \log p(\mathbf{W}|\mathbb{P}) + \log p(\boldsymbol{x}^{(t)}|\mathbf{W}, \mathbb{P}) - \log p(\boldsymbol{x}^{(t)}|\mathbb{P})$$

$$= \log \Gamma(\alpha_0) - \sum_{k=1}^{nC} \log \Gamma(\alpha_k) + \sum_{c=1}^{C} \sum_{i=1}^{n} (\alpha_{(c-1)n+i} - 1) \log w_{i,c}$$

$$+ \log \sum_{y_c \in \mathcal{Y}} p(x|y_c, \mathbf{W}, \mathbb{P})p(y_c|\mathbf{W}, \mathbb{P})$$

$$= \log \Gamma(\alpha_0) - \sum_{k=1}^{nC} \log \Gamma(\alpha_k) + \sum_{c=1}^{C} \sum_{i=1}^{n} (\alpha_{(c-1)n+i} - 1) \log w_{i,c}$$

$$+ \log \sum_{y_c \in \mathcal{Y}} \exp\left(\sum_{i=1}^{n} \left(w_{i,c}\boldsymbol{z}_{i,c}^{\mathrm{T}}\right)^{\top} \cdot \boldsymbol{z}^{\mathrm{I}}\right) \cdot \frac{\mathbb{1}_{\hat{y}_{ji}=y_c}}{\sum_{j'} \mathbb{1}_{\hat{y}_{j'i}=y_c}}$$

*Class-specific Dirichlet Prior.* We again start from the prior definition

$$p(W|\mathbb{P}) = \prod_{c=1}^{C} \mathrm{Dir}(W_c|\alpha_{c,1}, ..., \alpha_{c,n})$$

then turn into the log prior and posterior

$$\log p(\mathbf{W}|\mathbb{P}) = \sum_{c=1}^{C} \log \mathrm{Dir}(W_c|\alpha_{c,1}, ..., \alpha_{c,n})$$

$$= \sum_{c=1}^{C} \left[\log \Gamma(\alpha_{c,0}) - \sum_{i=1}^{n} \log \Gamma(\alpha_{c,i}) + \sum_{i=1}^{n} (\alpha_{c,i} - 1) \log w_{i,c}\right] \quad (\alpha_{c,0} = \sum_{i=1}^{n} \alpha_{c,i})$$

and log posterior

$$\log p(\mathbf{W}|\boldsymbol{x}^{(t)}, \mathbb{P}) = \sum_{c=1}^{C} \left[ \log \Gamma(\alpha_{c,0}) - \sum_{i=1}^{n} \log \Gamma(\alpha_{c,i}) + \sum_{i=1}^{n} (\alpha_{c,i} - 1) \log w_{i,c} \right]$$
$$+ \log \sum_{y_c \in \mathcal{Y}} p(x|y_c, \mathbf{W}, \mathbb{P}) p(y_c|\mathbf{W}, \mathbb{P})$$
$$= \sum_{c=1}^{C} \left[ \log \Gamma(\alpha_{c,0}) - \sum_{i=1}^{n} \log \Gamma(\alpha_{c,i}) + \sum_{i=1}^{n} (\alpha_{c,i} - 1) \log w_{i,c} \right]$$
$$+ \log \sum_{y_c \in \mathcal{Y}} \exp \left( \sum_{i=1}^{n} \left( w_{i,c} \boldsymbol{z}_{i,c}^{\mathrm{T}} \right)^{\top} \cdot \boldsymbol{z}^{\mathrm{I}} \right) \cdot \frac{\mathbb{1}_{\hat{y}_{ji}=y_c}}{\sum_{j'} \mathbb{1}_{\hat{y}_{j'i}=y_c}}$$

However, since Dirichlet priors would introduce additional steps (e.g., estimating concentration parameters $\alpha$), in our preliminary investigation, we used uniform prior to keep simplicity. Despite this simplest setup, our CARPRT prompt reweighting strategy effectively facilitated TTA methods. We leave more systematic explorations of alternative priors (e.g., Dirichlet) into future work.

## F.2 COMBINING CARPRT WITH SOFT PROMPT TUNING

*Soft prompt tuning* has recently become a powerful technique for adapting CLIP and other pre-trained vision-language models to downstream tasks. By learning optimal prompts that guide the model's understanding of new data, prompt tuning has shown remarkable effectiveness (Zhou et al., 2022b;a; Khattak et al., 2023b). ProDA optimizes prompt distributions to improve few-shot performance by training a set of learnable invisible prompt embeddings. While CARPRT is primarily designed to reweight visible prompt templates, our approach is not restricted to visible prompts. In this section, we also apply class-aware reweighting to the invisible prompts trained by ProDA, making our method capable of enhancing performance in various prompt tuning scenarios.

Our CARPRT method could enhance the ProDA framework by introducing a class-aware reweighting technique that adjusts the influence of each prompt based on the underlying class structure. Specifically, before each iteration of ProDA's prompt distribution learning, we use CARPRT to update the weights, which then guide the model's logit outputs for training the prompts. As the problem setting transitions from zero-shot to few-shot, our approach adapts by refining the weight estimation. Specifically, we use ground truth labels instead of the pseudo-labels for weight estimation, as shown in the following replacement for Eq. 10:

$$w'_{i,c} = \frac{\sum_{j=1}^{m} s_{j,i,c} \mathbb{1}_{y_j=y_c}}{\sum_{j=1}^{m} \mathbb{1}_{y_j=y_c}}, \tag{15}$$

where $y_j$ is the ground truth label of the sample $j$. The results shown in Table F.2 demonstrate that our method provides notable improvements in most datasets, highlighting the effectiveness of our class-aware prompt reweighting mechanism.

## F.3 COMBINING CARPRT WITH MODERN ZERO-SHOT METHODS

Recent zero-shot approaches often rely on large language models (LLMs) to generate class descriptions or prompts. While these methods have shown strong performance, they typically introduce external information and lack mechanisms to calibrate prompt relevance across classes. CARPRT can be applied on top of such methods to reweight their prompt pools in a class-aware manner, enhancing prediction quality without modifying the model or relying on additional supervision.

Beyond prompt-based methods, CARPRT is also compatible with image-centric approaches that construct classifiers directly from visual features, such as InMaP (Qian et al., 2024a). These two strategies are complementary: while InMaP builds a vision proxy via clustering, our method provides high-quality pseudo-labels that can guide its optimization. As shown in Tab. 6, integrating CARPRT with InMaP consistently improves performance. In particular, refining pseudo-labels using Sinkhorn

Table 5: Accuracy (%) comparison between our method and the *prompt tuning* baseline on fine-grained datasets using the CLIP-ViT-B/16 backbone. **Bold** values represent the highest accuracy in each row.

| | ProDA | ProDA + CARPRT |
|---|---|---|
| Caltech101 | 91.3 | **95.4** |
| DTD | **70.1** | 69.6 |
| EuroSAT | **84.3** | 83.4 |
| Aircraft | 36.6 | **36.9** |
| Food101 | 82.4 | **88.1** |
| Flower102 | 95.5 | **95.6** |
| Pets | 90.0 | **93.7** |
| Cars | 75.5 | **78.6** |
| Average | 78.2 | **80.2** |

distance leads to further gains, validating that better pseudo-labels directly reduce the theoretical gap between recovered and optimal vision proxies. These results highlight that CARPRT not only improves zero-shot inference on its own, but also serves as a valuable component within broader vision-language learning frameworks.

Table 6: Accuracy (%) comparison between our method and the baseline on ImageNet using the CLIP-ViT-B/16 and CLIP-ResNet50 backbone. **Bold** values represent the highest accuracy in each row.

| | InMaP | InMaP + CARPRT |
|---|---|---|
| CLIP-ViT-B/16 | | |
| w/o Skinhorn | 70.14 | **71.09** |
| Skinhorn | 72.55 | **72.57** |
| CLIP-ResNet50 | | |
| w/o Skinhorn | 60.83 | **60.95** |
| Skinhorn | **63.74** | 63.14 |

Table 7: Details for the datasets in our experiments.

| Dataset | Classes | Test Size |
|---|---|---|
| ImageNet | 1000 | 50,000 |
| Tiny-ImageNet | 200 | 10,000 |
| ImageNet-R | 200 | 30,000 |
| ImageNet-A | 200 | 6862 |
| ImageNet-Sketch | 1000 | 50,889 |
| ImageNet-V2 | 1000 | 10,000 |
| Caltech101 | 100 | 2465 |
| DTD | 47 | 1692 |
| EuroSAT | 10 | 8100 |
| Aircraft | 100 | 3333 |
| Food101 | 101 | 30,300 |
| Flowers102 | 102 | 2463 |
| Oxford Pets | 37 | 3669 |
| Cars196 | 196 | 8041 |
| Sun397 | 397 | 19,850 |
| UCF101 | 101 | 3783 |

Table 8: Accuracy (%) comparison between LLM-based prompt generation baselines and their combinations with our method on fine-grained datasets using the CLIP-ViT-B/16 backbone. **Bold** values represent the highest accuracy in each row.

| Method | Caltech101 | DTD | EuroSAT | Aircraft | Food101 | Flower102 | Pets | Cars | SUN397 | UCF101 | Average |
|---|---|---|---|---|---|---|---|---|---|---|---|
| CuPL | 93.68 | 50.27 | 52.69 | 25.57 | 86.71 | 71.31 | 89.10 | 65.31 | 65.13 | 70.33 | 67.01 |
| CuPL+Ours | 94.27 | 50.35 | 56.67 | 25.42 | 86.76 | 71.42 | **89.24** | 66.25 | 67.46 | 71.28 | 67.91 |
| MPVR | 93.98 | 50.12 | 55.47 | **26.18** | 86.89 | 72.14 | 89.07 | 66.97 | 65.24 | 70.42 | 67.65 |
| MPVR+Ours | 94.23 | 50.46 | **56.82** | 26.09 | **86.87** | **72.25** | **89.24** | 67.13 | 67.32 | **71.37** | **68.18** |
| VisDesc | **94.52** | **50.59** | 56.12 | 25.16 | 85.75 | 71.89 | 88.87 | **67.28** | **67.87** | 70.37 | 67.84 |

## F.4 COMBINING CARPRT WITH LLM-EMPOWERED PROMPT AUGMENTATION METHODS

Although CARPRT and LLM-empowered prompt augmentation methods are conceptually different, they can be combined in a complementary way. CARPRT is a training-free and inference-only method, relying solely on a fixed prompt template pool and without using any external knowledge such as LLMs. By contrast, CuPL (Shtedritski et al., 2023), MPVR (Mirza et al., 2024), and VisDesc (Menon & Vondrick, 2023b) generate class-specific prompts/descriptors via large language models and thus address a different setting. Importantly, these approaches are orthogonal to ours: while direct comparison is not the focus, CARPRT can reweight LLM-generated prompts, and combining them consistently brings further gains.

As shown in Tab. 8, integrating CARPRT with LLM-based prompt generation methods consistently improves their performance across datasets. This demonstrates that class-aware reweighting is complementary to LLM-generated prompts, enhancing their effectiveness without altering the underlying generation process. While VisDesc can be competitive or stronger in some cases, it requires a more complex pipeline and additional resources, whereas CARPRT provides a lightweight plug-in alternative.

## G ADDITIONAL EXPERIMENTS

### G.1 FILTERING LOW-CONFIDENCE PSEUDO-LABELS

We evaluate whether explicitly filtering low-confidence pseudo-labels improves performance. We consider two heuristics: (i) confidence-based filtering by thresholding the maximum similarity score, and (ii) entropy-based filtering by thresholding class-wise prediction entropy. As shown in Tab. 9, explicit filtering yields only marginal and inconsistent gains across datasets, is sensitive to the choice

Table 9: Accuracy (%) comparison between confidence-/entropy-based pseudo-label filtering variants and their combinations with CARPRT on fine-grained datasets. **Bold** values represent the highest accuracy in each row.

| Thresh. | Method | Aircraft | DTD | EuroSAT | Food101 | Pets | Caltech101 | Average |
|---|---|---|---|---|---|---|---|---|
| | | | | Confidence-based filtering (max score) | | | | |
| 0.30 | WPE | 22.28 | 47.18 | 52.37 | 85.49 | 81.46 | 92.62 | 63.57 |
| | CARPRT | 24.10 | 47.45 | 58.31 | 85.16 | 90.06 | 94.01 | 66.52 |
| | Filter Ratio | 0.51 | 0.15 | 0.16 | 0.53 | 0.60 | 0.34 | – |
| 0.25 | WPE | 22.11 | 47.18 | 51.88 | 85.34 | 80.92 | 94.24 | 63.61 |
| | CARPRT | 24.73 | 47.45 | 54.87 | 86.29 | 89.86 | 94.60 | 66.30 |
| | Filter Ratio | 0.98 | 0.77 | 0.77 | 0.97 | 0.98 | 0.81 | – |
| 0.00 (orig.) | WPE | 23.28 | 47.18 | 49.60 | 86.14 | 82.38 | 93.09 | 63.61 |
| | CARPRT | 24.49 | 48.90 | 55.56 | 86.31 | 89.13 | 94.16 | 66.43 |
| | | | | Entropy-based filtering (prediction entropy) | | | | |
| 2.0 | WPE | 21.90 | 44.80 | 51.92 | 85.41 | 92.57 | 63.11 | 63.11 |
| | CARPRT | 23.21 | 47.63 | 53.54 | 86.31 | 94.36 | 65.82 | 65.82 |
| | Filter Ratio | 0.18 | 0.41 | 0.88 | 0.91 | 0.88 | – | – |
| 2.5 | WPE | 21.95 | 45.85 | 49.60 | 85.33 | 92.61 | 62.81 | 62.81 |
| | CARPRT | 24.20 | 48.13 | 55.56 | 86.27 | 94.69 | 66.39 | 66.39 |
| | Filter Ratio | 0.38 | 0.62 | 1.00 | 0.97 | 0.92 | – | – |
| max (orig.) | WPE | 23.28 | 47.18 | 49.60 | 86.14 | 93.09 | 63.61 | 63.61 |
| | CARPRT | 24.49 | 48.90 | 55.56 | 86.31 | 94.16 | 66.43 | 66.43 |

of threshold, and can occasionally lead to performance drops. Overall, these results suggest that CARPRT is relatively robust to noisy pseudo-labels, and additional filtering heuristics provide limited practical benefit.

## G.2 DETAILED RESULTS FOR HYPERPARAMETER ANALYSIS

In this section, we analyze the impact of key hyperparameters across all fine-grained datasets, focusing on the temperature parameter $\tau$. In zero-shot classification, where only test data is available, conventional hyperparameter selection is inherently challenging due to the absence of training or validation data. Following Shu et al. (2018), we aim to identify hyperparameters that exhibit robust and consistent performance across diverse datasets.

As shown in Tab. 10, accuracy peaks at $\tau = 1.0$ and remains stable across a broad range, with a slight decline at higher values. A lower temperature, such as 0.5, sharpens focus on the most probable prompts but reduces distribution spread, limiting the ensemble effect of 247 prompt templates. This effect is crucial for capturing diverse information cues, and excessive concentration on dominant prompts may lead to performance degradation. While $\tau = 1.0$ may not be optimal for every dataset, it serves as a practical and generalizable choice under zero-shot constraints.

## G.3 RESULTS ON IMAGENET'S VARIANTS DATASETS

We also evaluate the performance of our method across Tiny-ImageNet and its variant datasets (ImageNet-A, ImageNet-R, ImageNet-Sketch, and ImageNet-V2), as shown in Tab. 11. The improvements on ImageNet and its variants datasets are smaller compared to those observed on the fine-grained datasets (shown in Tab. 1), for the following reasons. First, frequency bias is likely more pronounced in ImageNet and its variants. Given our use of a relatively small batch size of 512 and the exclusion of larger datasets such as LAION-400M for debiasing, the skewed class distribution may have negatively impacted the results. Second, the quality of the template pool plays a crucial role in model performance. According to (Allingham et al., 2023), the template pool was constructed by combining templates from 10 fine-grained datasets and 6 ImageNet and its variants datasets. Fine-grained datasets benefit more from the pool, as they can exploit class-specific templates. In contrast, the more diverse categories in ImageNet and its variants find less relevant information in the fine-grained templates, deriving less benefit from these templates. This mismatch reduces our

Table 10: Accuracy(%) results for varying temperature settings across fine-grained datasets using CLIP-ViT-B/16 and CLIP-ResNet50 backbones. **Bold** value represents the highest accuracy in each column.

| Temperature | Caltech101 | DTD | EuroSAT | Aircraft | Food101 | Flower102 | Pets | Cars | SUN397 | UCF101 | Average |
|---|---|---|---|---|---|---|---|---|---|---|---|
| | | | | | CLIP-ViT-B/16 | | | | | | |
| 0.5 | 93.45 | **49.13** | 53.29 | 23.97 | **87.26** | **71.82** | 88.69 | 64.66 | 66.32 | 69.68 | 66.83 |
| 1.0 (selected) | **94.16** | 48.90 | **55.56** | **24.49** | 86.31 | 71.36 | **89.13** | **66.14** | **66.93** | **70.41** | **67.34** |
| 2.0 | 94.07 | 48.54 | 55.19 | 24.17 | 85.87 | 71.12 | 88.69 | 65.67 | 66.07 | 70.11 | 66.95 |
| 3.0 | 93.93 | 48.27 | 55.15 | 24.04 | 85.74 | 70.95 | 88.39 | 65.29 | 65.98 | 70.09 | 66.78 |
| 4.0 | 93.87 | 48.16 | 55.07 | 23.96 | 85.69 | 70.93 | 88.36 | 65.21 | 65.91 | 69.95 | 66.71 |
| 5.0 | 93.72 | 48.09 | 54.92 | 23.87 | 85.62 | 70.85 | 88.31 | 65.14 | 65.88 | 69.77 | 66.62 |
| | | | | | CLIP-ResNet50 | | | | | | |
| 0.5 | **88.67** | 38.92 | 34.31 | 16.61 | **77.11** | **66.05** | **86.40** | 56.56 | 60.47 | 62.43 | 58.75 |
| 1.0 (selected) | 88.46 | 41.31 | **36.84** | **16.88** | 76.88 | 65.56 | 85.69 | 56.44 | **61.28** | **63.66** | **59.30** |
| 2.0 | 88.64 | 41.13 | 35.00 | 16.54 | 76.43 | 64.26 | 84.07 | **56.51** | 61.04 | 64.09 | 58.77 |
| 3.0 | 88.29 | **41.41** | 32.41 | 16.50 | 76.20 | 64.31 | 83.41 | 56.35 | 60.88 | 63.70 | 58.35 |
| 4.0 | 88.18 | 41.30 | 31.78 | 16.48 | 76.08 | 64.36 | 82.94 | 56.34 | 60.65 | 63.64 | 58.17 |
| 5.0 | 88.07 | 41.20 | 31.14 | 16.46 | 75.96 | 64.40 | 82.46 | 56.33 | 60.64 | 63.17 | 57.98 |

Table 11: Accuracy (%) comparison between baselines and our method on ImageNet and its variants using CLIP-ViT-B/16 and CLIP-ResNet50 backbones. **Bold** value represents the highest accuracy on each column. Standard deviations are shown inline using $\pm$.

| | ImageNet | Tiny-ImageNet | -A | -R | -Sketch | -V2 | Average |
|---|---|---|---|---|---|---|---|
| | | | CLIP-ViT-B/16 | | | | |
| MPE | 67.59 | 62.12 | 49.35 | 77.33 | 46.92 | 61.37 | 60.51 |
| WPE | 68.28$\pm$0.01 | 62.19$\pm$0.05 | 50.07$\pm$0.12 | 77.25$\pm$0.03 | 47.14$\pm$0.02 | 61.81$\pm$0.11 | 61.12$\pm$0.06 |
| CARPRT (Ours) | **68.59**$\pm$0.01 | **62.71**$\pm$0.04 | **51.60**$\pm$0.07 | **77.48**$\pm$0.04 | **47.53**$\pm$0.02 | **62.11**$\pm$0.09 | **61.67**$\pm$0.05 |
| | | | CLIP-ResNet50 | | | | |
| MPE | 59.12 | 43.32 | 46.25 | 69.05 | 39.05 | 54.05 | 53.50 |
| WPE | 59.78$\pm$0.01 | 43.12$\pm$0.08 | **46.37**$\pm$0.08 | 69.27$\pm$0.01 | 39.14$\pm$0.07 | 54.07$\pm$0.09 | 53.72$\pm$0.06 |
| CARPRT (Ours) | **59.98**$\pm$0.02 | **43.45**$\pm$0.06 | 46.19$\pm$0.09 | **69.59**$\pm$0.01 | **39.34**$\pm$0.04 | **54.26**$\pm$0.03 | **53.90**$\pm$0.06 |

method's effectiveness on ImageNet datasets, as it depends on template-provided information. These limitations suggest that mitigating frequency bias and enhancing template relevance for broader datasets could further improve CARPRT's performance.

### G.4 EXPERIMENTS ON IMBALANCED DATASETS

In this section, we evaluate the performance of CARPRT on datasets with class imbalances. Following Cao et al. (2019), we manually construct an imbalanced CIFAR-10 (Krizhevsky et al., 2009) dataset using an exponential decay strategy to create various degrees of class imbalance. We use an imbalance factor $\beta$ to describe the severity of the long-tailed distribution, defined as the ratio between the number of training samples in the most frequent class and the least frequent class. Specifically, $\beta$ is given by:

$$\beta = \frac{N_{\max}}{N_{\min}},$$

where $N_{\max}$ and $N_{\min}$ represent the number of training samples in the most frequent and least frequent classes, respectively. We conduct experiments with different imbalance ratios, setting $\beta = 10$, $\beta = 50$, and $\beta = 100$, using the CLIP-ViT-B/16 backbone.

The results shown in Tab. 12 demonstrate that CARPRT significantly outperforms the average baseline for all degrees of class imbalance. Specifically, CARPRT provides a consistent improvement in performance over WPE, though the gain decreases as the imbalance factor $\beta$ increases. This decreasing gain may be attributed to the global nature of the WPE weight estimation, which remains effective even under a higher imbalance. WPE calculates a single weight for the entire dataset, capturing the overall distribution and maintaining reasonable performance, even when certain classes are underrepresented.

Table 12: Accuracy (%) comparison between our method and baselines on CIFAR-10 using the CLIP-ViT-B/16 backbone. **Bold** values represent the highest accuracy in each column.

|  | Balanced Datasets | $\beta = 10$ | $\beta = 50$ | $\beta = 100$ |
|---|---|---|---|---|
| MPE | 89.56 | 89.58 | 89.57 | 89.56 |
| WPE | 89.55 | 90.02 | 90.78 | 91.07 |
| CARPRT (Ours) | **90.82** | **91.07** | **91.36** | **91.70** |

In contrast, CARPRT uses a per-class weighting strategy, which allows better adaptation to individual class characteristics, which is highly effective in balanced or moderately imbalanced settings. However, when the class imbalance becomes severe, the challenge arises for classes with very few samples (e.g., only 10 samples). In these cases, the reliability of CARPRT's weight estimates decreases as a result of insufficient data, impacting performance.

### G.5 IMPACT OF TEMPLATE QUALITY

In this section, we investigate the impact of template quality on ImageNet classification tasks. Specifically, we explore how different prompt template pools influence performance by evaluating two newly generated template pools alongside the original templates on the ImageNet datasets. Specifically, Pool1 was generated using Claude 3.5 (Anthropic, 2024) to produce 300 templates tailored to the ImageNet label space. Each category in Pool1 consists of 100 prompt templates structured in descriptive formats, such as *"A photo of a ", "A photo of a ", "The type of "*. These templates aim to incorporate task-specific context and improve the alignment between the prompts and ImageNet categories. Pool2, on the other hand, was constructed using Phi 3.1 (Microsoft, 2024) to create highly descriptive templates. For each ImageNet category, Phi 3.1 generated five detailed prompts, resulting in a total of 5,000 templates across all categories. These templates focus on providing class-specific descriptive information, enabling a more precise and nuanced interaction with the underlying vision-language model. These additional template pools were evaluated on ImageNet dataset compared to the original templates (Pool0), as shown in Tab. 13.

Table 13: Accuracy (%) comparison across different template pools using WPE and CARPRT methods on ImageNet classification.

| Pool | Method | ImageNet Acc. (%) | Perf. Comparison |
|---|---|---|---|
| Pool0 | WPE | 68.28 | – |
|  | CARPRT | **68.59** | +0.31 |
| Pool1 | WPE | 68.35 | – |
|  | CARPRT | **68.61** | +0.26 |
| Pool2 | WPE | 68.34 | – |
|  | CARPRT | **68.97** | +0.63 |

Pool1 targets more task-specific information by generating templates with respect to the ImageNet label space. This leads to performance improvements for both WPE and CARPRT prompt reweighting strategies compared to Pool0. On the other hand, the generated templates in Pool2 incorporate more class-specific descriptive information. CARPRT benefits significantly from these templates, achieving greater performance gains compared to WPE. This highlights the effectiveness of class-aware prompt reweighting in leveraging descriptive templates.

**Future Work**. Results in App. G.5 show that a high-quality prompt template pool significantly improves performance. Building on these results and the previously discussed limitations, a key direction for future work is enhancing the quality and diversity of the prompt template pool, which existing methods often overlook. Future research could focus on cost-effective strategies for generating and evaluating diverse, representative prompts. This may include developing metrics to assess how well prompts capture class-specific characteristics and enhancing inter-class distinctions to improve the model's ability to differentiate closely related categories.

Table 14: Comparison of normalization schemes under WPE and CARPRT. Accuracy (%) is reported on Fine-Grained, ImageNet, and Variant subsets, along with the average across them.

| Method | Normalization Schemes | Fine-Grained | ImageNet | Variant | Average |
|--------|----------------------|--------------|----------|---------|---------|
| WPE | none | 64.82 | 68.28 | 59.69 | 64.26 |
| | test | 64.93 | 68.45 | 59.72 | 64.37 |
| | pre-train | **65.01** | **68.64** | 59.57 | 64.41 |
| | both | 65.00 | 68.56 | **59.74** | **64.43** |
| CARPRT | none | 67.34 | 68.59 | 60.39 | 65.44 |
| | test | 67.12 | 68.27 | 60.18 | 65.19 |
| | pre-train | **67.45** | 68.72 | **60.55** | 65.57 |
| | both | 67.44 | **68.77** | 60.53 | **65.58** |

### G.6 ANALYSIS OF FREQUENCY BIAS CORRECTION

To correct potential biases introduced by the class frequency distribution in the pre-training or test-time datasets, Allingham et al. (2023) applies normalization to the score matrix before computing the prompt weights. This step ensures that the scale and distribution of class-prompt scores are consistent across categories and prompts, thereby mitigating dataset-specific artifacts that could affect final predictions. The scores $s_{j,i,c}$ across all images $\boldsymbol{x}_j$ predicted to class $y_c$ under prompt $p_i$ are normalized as follows:

$$\tilde{s}_{j,i,c} = s_{j,i,c} - \mu, \tag{16}$$

where $\mu$ denotes the mean of the scores, computed differently depending on the normalization scheme: (1) **none**: No normalization is applied and we set $\mu = 0$; (2) **test**: $\mu$ is computed by the test data scores: $\mu = \mu^{\text{test}} = \frac{1}{N^{\text{test}}} \sum_{j=1}^{N^{\text{test}}} s_{j,i,c}$; (3) **pre-train**: $\mu$ is computed by the data drawn from LAION-400m (Schuhmann et al., 2021), following Allingham et al. (2023): $\mu = \mu^{\text{pre}} = \frac{1}{N^{\text{pre}}} \sum_{j=1}^{N^{\text{pre}}} s_{j,i,c}$; (4) **both**: Combine the two sources by interpolation: $\mu = (\mu^{\text{test}} + \mu^{\text{pre}})/2$. These normalized scores are then used to compute prompt weights.

As shown in Tab. 14, the WPE method benefits noticeably from normalization. All normalization schemes improve over the unnormalized baseline, with the `both` setting achieving the best overall performance. This suggests that WPE is sensitive to distributional bias and gains from explicitly correcting both pre-training and test-time frequency effects.

By contrast, CARPRT performs robustly across all settings. Even without normalization, CARPRT outperforms WPE, and gains only slight improvements from applying `pre-train` or `both` normalization. Interestingly, `test`-only normalization slightly reduces performance, indicating that test-derived statistics may inject noise rather than correct meaningful bias. This robustness likely stems from the class-aware formulation of CARPRT, which captures prompt-class dependencies more explicitly.

In summary, while WPE requires normalization to mitigate its reliance on biased score distributions, CARPRT consistently maintains strong performance, demonstrating its effectiveness as a prompt reweighting method.

## H DISCUSSION OF PRIOR DISTRIBUTION OF THE PROMPT WEIGHTS $\Pr(\mathbf{W}|\mathbb{P})$

We extend the discussion of the proposed probabilistic interpretation (Sec. 3) to the weights prior $\Pr(\mathbf{W}|\mathbb{P})$. In the current zero-shot classification scenario addressed by CARPRT, there is no optimization-based process for "estimating" the weights, and as such, the weight prior $\Pr(\mathbf{W}|\mathbb{P})$ does not play a role in the methodology. Nevertheless, our probabilistic framework is flexible enough to accommodate more general trainable settings, such as active learning and few-shot estimation, where the probabilistic formulation becomes particularly beneficial. In these cases, a discussion of

the weight prior would provide valuable insights and contribute to a more complete understanding of the framework's advantages.

Suppose there is a label space $\mathcal{Y}$ with size $|\mathcal{Y}| = C$. Let $\mathbb{P} = \{p_i\}_{i=1}^n$ be a pool of $n$ independent prompt templates. Let $\mathbf{W} = \{\mathbf{W}_c\}_{c=1}^C$ be our weight matrix. Recall that $\mathbf{W}_c \in \Delta^{n-1}$ is the $(n-1)$-dimensional probability simplex, representing the weights for class $y_c$ across all prompts.

We consider three choices of priors: uniform prior, global Dirichlet prior, and class-specific Dirichlet priors.

**Uniform Prior**. The uniform prior assumes all valid weight configurations are equally likely a priori.

$$p(\mathbf{W}|\mathbb{P}) = \begin{cases} \frac{1}{|\mathcal{W}|} & \text{if } \mathbf{W} \in \mathcal{W} \\ 0 & \text{otherwise} \end{cases}$$

where $\mathcal{W} = \{\mathbf{W} \in \mathbb{R}^{n \times C} : W_c \in \Delta^{n-1} \text{ for all } c \in \{1, ..., C\}\}$.

The uniform prior is the easiest setup to implement and does not introduce bias towards any particular weight configuration. However, the uniform prior does not leverage any prior knowledge about the prompts, which is prone to overfitting with limited data (when adapted to trainable setting).

**Global Dirichlet Prior**. This defines a single Dirichlet distribution over all weights, treating them as a single vector.

$$p(\mathbf{W}|\mathbb{P}) = \text{Dir}(\text{vec}(\mathbf{W})|\alpha_1, ..., \alpha_{nC})$$

where $\text{vec}(\mathbf{W})$ is the vectorization of $\mathbf{W}$, and $\alpha_i > 0$ are concentration parameters of the Dirichlet distribution.

Compared to uniform prior, Dirichlet prior can encode varying degrees of certainty about different weights. Moreover, it is conjugate to multinomial likelihood, allowing for closed-form posterior updates for certain model setup. This can also align with WPE-like class-shared-weighting strategies. However, it ignores the class structure and treats all weights as part of a single distribution, potentially missing class-specific patterns.

**Class-specific Dirichlet Prior**. This strategy sets an independent Dirichlet distribution for each class's weight, and stacks a product of $C$ classes' Dirichlet distributions.

$$p(\mathbf{W}|\mathbb{P}) = \prod_{c=1}^C \text{Dir}(\mathbf{W}_c|\alpha_{c,1}, ..., \alpha_{c,n})$$

where $\alpha_{c,i} > 0$ are class and prompt-specific contenration parameters.

Currently, this setup best suits our class-aware prompt reweighting mechanism, as it allows for different prior beliefs about weight distributions for each class, class-specific modeling. Compared with global Dirichlet, it reduces dimensionality - each Dirichlet distribution is over $n$ parameters, not $n \times C$ anymore. More importantly, it aligns with the per-class simplex constraint of the weight space.

**Entropy Analysis**. Different prior choices lead to different entropy results. The uniform prior has an associated entropy as

$$H[p(\mathbf{W}|\mathbb{P})]_{\text{uniform}} = \log |\mathcal{W}|,$$

where $|\mathcal{W}|$ is the volume of the weight space.

As for global Dirichlet prior, we have

$$H[p(\mathbf{W}|\mathbb{P})] = \log B(\alpha) + (\alpha_0 - nC)\psi(\alpha_0) - \sum_{i=1}^{nC}(\alpha_i - 1)\psi(\alpha_i),$$

where $B(\cdot)$ is the multivariate beta function, and $\psi(\cdot)$ is the digamma function.

The entropy for class-specific Dirichlet priors is

$$H[p(\mathbf{W}|\mathbb{P})] = \sum_{c=1}^C \left(\log B(\alpha_c) + (\alpha_{c,0} - n)\psi(\alpha_{c,0}) - \sum_{i=1}^n(\alpha_{c,i} - 1)\psi(\alpha_{c,i})\right),$$

where $\alpha_c = (\alpha_{c,1}, ..., \alpha_{c,n})$ and $\alpha_{c,0} = \sum_{i=1}^n \alpha_{c,i}$ for each class $c$.

When we are setting the equal concentration parameters, such that $\alpha_i = \alpha$ for all $i$ in the global Dirichlet, and $\alpha_{c,i} = \alpha$ for all $c, i$ in the class-specific Dirichlets, and let $\alpha = 1$, the uniform prior has the highest entropy (uninformative), while the class-specific Dirichlets having the lowest entropy. This is because the class-specific Dirichlets with $\alpha = 1$ are equivalent to independent uniform distributions over smaller simplices, further concentrating the probability.

## I    DETAILED PROOFS

**Lemma 2** (Relative Likelihood *cf.* Lemma 1). *The likelihood of an image $\boldsymbol{x}$, given class c, prompt weights $\mathbf{W}$ and a prompt pool $\mathbb{P}$, following the EBM defined in Eq. 6, is proportional to:*

$$\Pr(\boldsymbol{x}_j|y_c, \mathbf{W}, \mathbb{P}) \propto \exp\left\{ sim(\boldsymbol{z}_j^{\mathrm{I}}, \boldsymbol{z}_c^{\mathrm{T}}) \right\} \propto \exp\left\{ \sum_{i=1}^n (w_{i,c}\, \boldsymbol{z}_{i,c}^{\mathrm{T}})^\top \cdot \boldsymbol{z}^{\mathrm{I}} \right\}, \qquad (17)$$

*where $\boldsymbol{z}_j^{\mathrm{I}} = f(\boldsymbol{x}_j)$ and $\boldsymbol{z}_{i,c}^{\mathrm{T}} = g(p_i(y_c))$ are image embeddings of sample $\boldsymbol{x}_j$ and text embeddings of class $y_c$ under prompt $p_i$, respectively.*

*Proof.* **Similarity as Negative Energy**. As with (LeCun et al., 2006), a general form of EBMs is given by $P_\theta(x) = \exp(-\beta E_\theta(x))/Z(\theta)$, which enables us to define unnormalized energy function with a partition function for normalization. Therefore, in our zero-shot classification context, we define the energy function with respect to the score function of the CLIP.

$$E(\boldsymbol{x}_j, y_c, \mathbf{W}, \mathbb{P}) = \mathrm{sim}(\boldsymbol{z}_j^{\mathrm{I}}, \boldsymbol{x}_c^{\mathrm{T}})$$

This score function measures the compatibility between the image embedding $\boldsymbol{z}_j^{\mathrm{I}}$ and the text embedding embedding $\boldsymbol{x}_c^{\mathrm{T}}$ of class $y_c$. higher compatibility corresponds to lower energy, aligning with the EBM principle that more likely configurations (of model) have lower energy.

**Intractable Partition Function**. Computing the partition function is intractable since we need to marginalize over the image space. However, what we care about is the relative relation between $\Pr(\boldsymbol{x}_j|y_c, \mathbf{W}, \mathbb{P})$ and $\Pr(\boldsymbol{x}_j|y_{c'}, \mathbf{W}, \mathbb{P})$, we can safely drop off the partition function in our relative likelihood.

**Similarity Computation**. Consider a general linear combination of similarities for a prompt ensemble:

$$\mathrm{sim}(\boldsymbol{z}^{\mathrm{I}}, \boldsymbol{z}_c^{\mathrm{T}}) = h_c \left( \{\mathrm{sim}(\boldsymbol{z}^{\mathrm{I}}, \boldsymbol{z}_{i,c}^{\mathrm{T}})\}_{i=1}^n \right)$$

$$h_c(\{s_i\}_{i=1}^n) = \sum_{i=1}^n \alpha_{i,c} s_i + \beta_c$$

where $h_c : \mathbb{R}^d \to \mathbb{R}$ is a function that linearly combines the similarities over all prompts $p_i \in \mathbb{P}$ for a specific class $y_c$. $\alpha_{i,c} \in \mathbb{R}$ and $\beta_c \in \mathbb{R}$ are weights and bias terms. Substituting $s_i = \mathrm{sim}(\boldsymbol{z}^{\mathrm{I}}, \boldsymbol{z}_{i,c}^{\mathrm{T}}) = \boldsymbol{z}_{i,c}^{\mathrm{T}\top} \cdot \boldsymbol{z}^{\mathrm{I}}$, we get:

$$\mathrm{sim}(\boldsymbol{z}_j^{\mathrm{I}}, \boldsymbol{z}_{i,c}^{\mathrm{T}}) = \sum_{i=1}^n \alpha_{i,c} (\boldsymbol{z}_{i,c}^{\mathrm{T}})^\top \cdot \boldsymbol{z}_j^{\mathrm{I}} + \beta_c$$

We can then absorb the bias term $\beta_c$ into the exponential function,

$$\Pr(\boldsymbol{x}_j|y_c, \mathbf{W}, \mathbb{P}) \propto \exp(\mathrm{sim}(\boldsymbol{z}_j^{\mathrm{I}}, \boldsymbol{z}_{i,c}^{\mathrm{T}}))$$

$$= \exp(\sum_{i=1}^n \alpha_{i,c} (\boldsymbol{z}_{i,c}^{\mathrm{T}})^\top \cdot \boldsymbol{z}_j^{\mathrm{I}} + \beta_c)$$

$$= \exp(\beta_c) \exp(\sum_{i=1}^n \alpha_{i,c} (\boldsymbol{z}_{i,c}^{\mathrm{T}})^\top \cdot \boldsymbol{z}_j^{\mathrm{I}})$$

$$\propto \exp(\sum_{i=1}^n (\alpha_{i,c} \boldsymbol{z}_{i,c}^{\mathrm{T}})^\top \cdot \boldsymbol{z}_j^{\mathrm{I}}).$$

By setting $w_{i,c} = \alpha_{i,c}$, we arrive at the formulation in Lemma 1. $\qquad\square$

**Proposition 3** (*cf.* Proposition 2). *Let $\mathcal{X}$ be the image space, $\mathcal{Y}$ be the class space. Given a set of prompts $\mathbb{P}$, for any prompt weighting scheme $S$ (cf. Eq. 1), define the representable likelihood set $\mathcal{F}_S$ as:*

$$\mathcal{F}_S = \{f : \mathcal{X} \times \mathcal{Y} \to \mathbb{R}_+ | \exists \mathbf{W} \in \mathcal{W}_S, \mathbb{P}, \text{ s.t. } f(\boldsymbol{x}, y_c) \propto \Pr(\boldsymbol{x}|y_c, \mathbf{W}, \mathbb{P})\},$$

*where $\mathcal{W}_S$ is the weight space under the scheme $S$. Let $\mathcal{F}_{CI}$ and $\mathcal{F}_{CS}$ be the representable likelihood set induced from class-independent weighting and class-aware weighting (cf. Eq. 1) schemes. Then, we have: $\exists f^* \in \mathcal{F}_{CS}$ such that $\forall f_{CI} \in \mathcal{F}_{CI}, \exists \boldsymbol{x} \in \mathcal{X}, y_c \in \mathcal{Y}$ where $f^*(\boldsymbol{x}, y_c) \neq f_{CI}(\boldsymbol{x}, y_c)$.*

*Proof.* We prove this by constructing a specific function in $\mathcal{F}_{CS}$ and showing it cannot be represented by any function in $\mathcal{F}_{CI}$. For simplicity, we consider a **toy** setting with three classes $\mathcal{Y} = \{y_1, y_2, y_3\}$ and two prompts $\mathbb{P} = \{p_1, p_2\}$. For any $\boldsymbol{x} \in \mathcal{X}$, the function under class-aware weighting for $\forall y_c \in \{y_1, y_2, y_3\}$ takes the form:

$$f^*(\boldsymbol{x}, y_c) = \sum_{i=1}^{|\mathbb{P}|} w_{i,c} \Pr(\boldsymbol{x}|y_c, p_i)$$
$$= w_{1,c} \Pr(\boldsymbol{x}|y_c, p_1) + w_{2,c} \Pr(\boldsymbol{x}|y_c, p_2).$$

where $w_{i,j} \in \mathbb{R}_+$ are class-aware weights for prompt $i$ and class $j$. For ease of notation, we denote the prompt-conditional likelihood by $a_{i,c} \triangleq \Pr(\boldsymbol{x}|y_c, p_i)$. This way $f^* \in \mathcal{F}_{CS}$ can be expressed as

$$f^*(\boldsymbol{x}, y_1) = w_{1,1}a_{1,1} + w_{2,1}a_{2,1}$$
$$f^*(\boldsymbol{x}, y_2) = w_{1,2}a_{1,2} + w_{2,2}a_{2,2}$$
$$f^*(\boldsymbol{x}, y_3) = w_{1,3}a_{1,3} + w_{2,3}a_{2,3}$$

We then consider a specific instance[4] of this function by choosing:

$$w_{1,1} = 2, \quad w_{2,1} = 1$$
$$w_{1,2} = 1, \quad w_{2,2} = 2$$
$$w_{1,3} = 3, \quad w_{2,3} = 3$$

This leads to

$$f^*(\boldsymbol{x}, y_1) = 2a_{1,1} + a_{2,1}$$
$$f^*(\boldsymbol{x}, y_2) = a_{1,2} + 2a_{2,2}$$
$$f^*(\boldsymbol{x}, y_3) = 3a_{1,3} + 3a_{2,3}$$

Now, suppose for contradiction that $\exists f_{CI} \in \mathcal{F}_{CI}$ such that $f^* = f_{CI}$. By definition of $\mathcal{F}_{CI}$, $f_{CI}$ takes the form $f_{CI}(\boldsymbol{x}, y_c) = w_1 a_{1,c} + w_2 a_{2,c}$, where $w_1, w_2 \in \mathbb{R}_+$ are class-independent weights.

If $f^* = f_{CI}$, then for all classes $y_c \in \{y_1, y_2, y_3\}$, we must have the following equations to hold simultaneously:

$$2a_{1,1} + a_{2,1} = w_1 a_{1,1} + w_2 a_{2,1} \quad \text{(for } y_1\text{)}$$
$$a_{1,2} + a_{2,2} = w_1 a_{1,2} + w_2 a_{2,2} \quad \text{(for } y_2\text{)}$$
$$3a_{1,3} + 3a_{2,3} = w_1 a_{1,3} + w_2 a_{2,3} \quad \text{(for } y_3\text{)}$$

From these equations, we can deduce that

$$w_1 = 2 \text{ and } w_2 = 1 \text{ must hold for any } a_{1,1}, a_{2,1} > 0 \quad \text{(for } y_1\text{)}$$
$$w_1 = 1 \text{ and } w_2 = 2 \text{ must hold for any } a_{1,2}, a_{2,2} > 0 \quad \text{(for } y_2\text{)}$$
$$w_1 = 3 \text{ and } w_2 = 3 \text{ must hold for any } a_{1,3}, a_{2,3} > 0 \quad \text{(for } y_1\text{)}$$

Thus, we need $w_1 = 2$ for $y_1$ while $w_1 = 1$ for $y_2$, immediately leading to a contradiction as $w_1$ cannot simultaneously equal 1 and 2.

Therefore, no class-independent weighting scheme can represent the function $f^*$ we constructed. We have proven that $\exists f^* \in \mathcal{F}_{CS}$ such that $\forall f_{CI} \in \mathcal{F}_{CI}, \exists \boldsymbol{x} \in \mathcal{X}, y_c \in gY$ where $f^*(\boldsymbol{x}, y_c) \neq f_{CI}(\boldsymbol{x}, y_c)$. □

---

[4]unnormalized weights, just for illustration

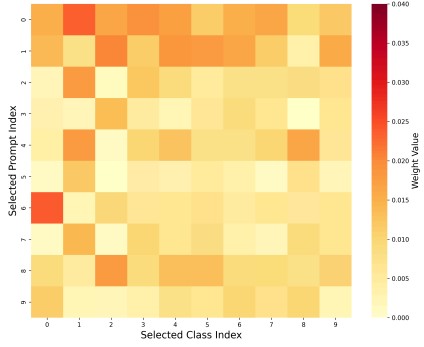 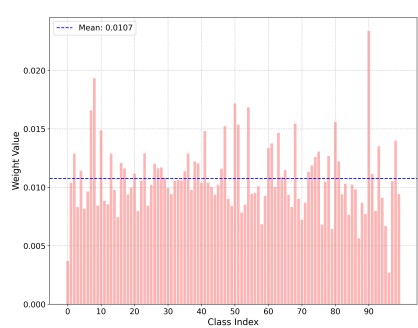

(a) Truncated Class-Prompt Weights Heatmap

(b) Per-Class Weight Distribution for "*a low resolution photo of a* {}."

Figure 6: Visualization of the class-aware prompt weights estimated by CARPRT on the Caltech101 dataset. (a) The heatmap shows the prompt weights across a subset of classes and prompts, revealing diverse weight patterns and confirming class-specific preferences. (b) The bar plot displays the distribution of prompt weights assigned to the prompt "*a low resolution photo of a* {}" across all classes.

## J    ADDITIONAL VISUALIZATIONS OF PROMPT WEIGHTS

To provide qualitative insight into CARPRT's mechanism, we first visualize the learned class-specific prompt weights on the *Caltech101* dataset. Fig. 6(a) shows the *truncated* weight matrix for a subset of prompts ($n' < n$ columns) and classes ($C' < C$ rows) from the full matrix $\mathbf{W} \in \mathbb{R}^{n \times C}$, where clear differences in the weights assigned to the same prompt across different classes are evident. Fig. 6(b) further illustrates this class-dependency by plotting the weights of a single prompt template—"*a low resolution photo of a* {}"—across all classes, demonstrating that the contribution of this prompt is tailored to each class. These visualizations corroborate our quantitative results, confirming that CARPRT prioritizes prompts differently for each class.

In addition, we include additional visualizations of the CARPRT-generated prompt weights across all ten fine-grained datasets in the supplementary material (due to file size, these figures are not embedded in the main PDF). Each visualization is presented as a heatmap, where the vertical axis corresponds to the prompt index and the horizontal axis to the class index.

These heatmaps consistently reveal the class-specific nature of the learned weights: the columns exhibit noticeable variation across prompts rather than remaining uniform, indicating that different prompts are emphasized for different classes. Moreover, for most fine-grained datasets, only a small subset of prompts receive high weights across classes, while the majority are down-weighted—this sparsity manifests visually as a few strong horizontal lines. This trend is particularly evident on `Food101`, where the semantic homogeneity of the dataset leads to more consistent prompt preferences across classes.

Nevertheless, even within `Food101`, the highest-weighted prompt still varies across classes, demonstrating that class-aware prompt weighting remains essential. These results collectively support the effectiveness of WPE (Allingham et al., 2023) in highlighting useful prompts for the dataset, while also confirming the necessity of CARPRT's class-aware weighting to fully capture intra-dataset variation.

## K    USE OF LARGE LANGUAGE MODELS (LLMS)

In preparing this submission, we used LLMs solely as writing aids to improve readability. Specifically, LLMs were employed to correct grammar errors and polish the text. No part of the scientific content—including problem formulation, method design, experiments, or analysis—is generated by LLMs. All technical contributions and claims were conceived, implemented, and evaluated by the authors.

