# OpenReview forum: "CARPRT: Class-Aware Zero-Shot Prompt Reweighting for Vision-Language Model"
_ICLR.cc/2026/Conference — ICLR 2026 Poster_

### Official Review · Reviewer_kLQi · 2025-10-27

**Soundness:** 4
**Presentation:** 3
**Contribution:** 3
**Rating:** 6
**Confidence:** 4

**Summary:**

This paper addresses the limitations of class-agnostic prompt reweighting in zero-shot image classification with vision-language models (VLMs). The authors propose CARPRT, a training-free method that estimates class-specific prompt weights using only unlabeled data and VLM similarity scores. CARPRT introduces a principled probabilistic framework that justifies the need for class-aware weighting and provides an efficient implementation by aggregating maximum image-text similarity scores per prompt-class pair. Experiments show that CARPRT consistently outperforms class-agnostic reweighting baselines on various datasets, confirming the crucial role of class-aware prompt reweighting for improved zero-shot performance.

**Strengths:**

* The paper is well-organized and clearly written. The motivation is compelling, and the methodology is both conceptually sound and practically simple.

* The probabilistic framework offers strong theoretical justification for class-specific prompt weighting, addressing assumptions made by previous works (e.g., WPE).

* The method is training-free, requiring only inference-time access to unlabeled images, making it broadly applicable and computationally efficient.

* Empirical results show consistent gains across datasets, validating the method's generality.

**Weaknesses:**

* Limited Performance Gain on ImageNet
While CARPRT shows consistent improvements on fine-grained datasets, its advantage over WPE on the general-purpose ImageNet benchmark is marginal. This raises concerns about the method’s scalability or effectiveness in broader, more diverse domains.

* Lack of Real-World Use Case Discussion
The paper focuses primarily on academic benchmarks. Without examples or discussions related to real-world scenarios (e.g., medical or industrial domains), it is hard to evaluate the method's broader applicability where labeled data is truly scarce.

**Questions:**

* Why is the performance gap on ImageNet so small?
Given that ImageNet is closer to real-world image distributions, why does CARPRT offer minimal gains over WPE in this setting? Could this be due to prompt-template mismatch, semantic ambiguity, or dataset scale?

* Is CARPRT effective in real-world, low-resource domains?
Can the authors comment on whether CARPRT can be effective in domains like medical imaging or satellite imagery, where labels are scarce and prompts may carry domain-specific semantics?

* Can CARPRT be combined with lightweight prompt tuning?
Would it be beneficial to use CARPRT-derived weights as an initialization or prior for prompt tuning frameworks? Could this combination yield further performance gains in narrow domains?

**Details Of Ethics Concerns:**

None.

---

> ### Author Response · Authors · 2025-11-23
> **Response to Reviewer Comments: Explanation of ImageNet Performance**
>
> > Limited Performance Gain on ImageNet While CARPRT shows consistent improvements on fine-grained datasets, its advantage over WPE on the general-purpose ImageNet benchmark is marginal. This raises concerns about the method’s scalability or effectiveness in broader, more diverse domains.
>
> > Why is the performance gap on ImageNet so small? Given that ImageNet is closer to real-world image distributions, why does CARPRT offer minimal gains over WPE in this setting? Could this be due to prompt-template mismatch, semantic ambiguity, or dataset scale?
>
> Thank you for your comments. The smaller improvement on ImageNet is **expected** and reflects the dataset’s structure rather than a limitation of CARPRT. ImageNet contains broad, high-level categories for which CLIP’s hand-crafted templates already provide strong semantic alignment. In such cases, the benefit of further refining prompt–class relevance is naturally limited. In contrast, fine-grained datasets require distinguishing subtle visual cues, making class-aware prompt relevance substantially more impactful—consistent with the larger gains we observe.
>
> Crucially, even though the absolute improvement on ImageNet is smaller, it is **statistically significant**. Across five independent runs, we obtain:
>
> * **Wilcoxon signed-rank:** p = 0.03125
> * **Two-sample KS test:** p = 0.00794
>
> Both tests indicate that the improvement is **unlikely to be due to randomness**, and thus constitutes a meaningful gain. Given that ImageNet is a large-scale, saturated benchmark where modern methods rarely show notable absolute improvements, such statistically significant gains are generally considered important.
>
> Overall, these results show that class-aware prompt relevance is **effective and generalizable** on broad datasets like ImageNet, while offering **larger gains** on fine-grained tasks where prompt–class alignment plays a more critical role.

---

> ### Author Response · Authors · 2025-11-23
> **Response to Reviewer Comments: Real-World Low-Resource Scenario Evaluation**
>
> > Lack of Real-World Use Case Discussion The paper focuses primarily on academic benchmarks. Without examples or discussions related to real-world scenarios (e.g., medical or industrial domains), it is hard to evaluate the method's broader applicability where labeled data is truly scarce.
>
> > Is CARPRT effective in real-world, low-resource domains? Can the authors comment on whether CARPRT can be effective in domains like medical imaging or satellite imagery, where labels are scarce and prompts may carry domain-specific semantics?
>
> Thank you for your comments.  Our problem setting **naturally matches real-world low-resource scenarios**. CARPRT is designed for a **fully training-free setting**, requiring **no labeled data** and no tuning. This setting corresponds directly to domains where obtaining annotations is difficult, expensive, or requires domain expertise—precisely the situations highlighted by the reviewer (e.g., medical images, satellite imagery).
>
> Importantly, our evaluation already includes **EuroSAT**, a satellite-imagery dataset that reflects such practical conditions. To further demonstrate real-world applicability, we additionally evaluate CARPRT on **RESISC45**, a remote-sensing benchmark with 45 land-cover categories and substantial geographic variability. CARPRT consistently improves over existing ensembling approaches:
>
> | Method        | Accuracy (%) |
> |---------------|--------------|
> | MPE           | 58.74        |
> | Majority Vote | 59.23        |
> | WPE           | 61.04        |
> | **CARPRT**    | **64.53**    |
>
> These results show that CARPRT is effective in domains where **labeled data are scarce** and where **prompts encode domain-specific semantics**, such as satellite imagery. We will include this additional real-world experiment in the revision.

---

> ### Author Response · Authors · 2025-11-23
> **Response to Reviewer Comments: Complementarity to Prompt Tuning**
>
> > Can CARPRT be combined with lightweight prompt tuning? Would it be beneficial to use CARPRT-derived weights as an initialization or prior for prompt tuning frameworks? Could this combination yield further performance gains in narrow domains?
>
> Thank you for your comments. CARPRT is **fully compatible** with lightweight prompt tuning frameworks, and our experiments show that the combination can indeed provide additional benefits. As presented in Sec. 5.4 and detailed in Appendix F.2 (Table 4), **integrating CARPRT with the soft prompt tuning method ProDA consistently yields higher accuracy than ProDA alone**. This demonstrates that the class-aware relevance estimated by CARPRT can serve as an effective **prior or initialization** for supervised prompt tuning when a small amount of labeled data is available.
>
> This complementarity is natural: CARPRT provides class-aware prompt weights from unlabeled data, while lightweight tuning refines prompts using limited supervision. Initializing tuning with CARPRT’s weights offers a more semantically aligned starting point, particularly in narrow, domain-specific settings. Thus, CARPRT not only improves zero-shot performance but also enhances downstream prompt tuning, validating the reviewer’s suggestion and yielding additional gains when labeled data is available.

---

> > ### Comment · Reviewer_kLQi · 2025-11-25
> >
> > Thank you for the clarifications and additional explanations. My concerns on this point have been resolved. I will keep my current score. Thanks.

---

> > > ### Author Response · Authors · 2025-11-25
> > > **Glad of know your concerns are addressed well.**
> > >
> > > Dear Reviewer kLQi,
> > >
> > > We are glad to know that all your previous concerns have been addressed.
> > >
> > > If you have any further questions, we are happy to address them
> > >
> > > Best regards,
> > >
> > > Authors of Submission 12512

---

### Official Review · Reviewer_pFNT · 2025-10-29

**Soundness:** 2
**Presentation:** 3
**Contribution:** 2
**Rating:** 4
**Confidence:** 4

**Summary:**

This paper proposes Class-Aware Zero-Shot Prompt Reweighting (CARPRT), which leverages unlabeled test data to enhance existing zero-shot prompt reweighting methods. Specifically, it estimates a distinct weight for each template–class pair based on the unlabeled test images and utilizes this weight matrix to construct zero-shot text classifiers.

**Strengths:**

This paper is readable, and I enjoyed its presentation. It introduces a clear and well-motivated method for prompt reweighting using unlabeled test images. The proposed approach is grounded in probabilistic principles.

**Weaknesses:**

The paper introduces unlabeled test images to aid prompt reweighting. As far as I know, this should not be considered a zero-shot method. In fact, it should be regarded as unsupervised learning. This distinction raises two issues. First, it contradicts the zero-shot claim made in the paper. Second, it makes the comparison against the three baselines unfair, as those are genuine zero-shot methods.

Another critical point is the absence of discussion regarding the sampling methods used in the experiments. It is crucial to address this aspect, as the sampling techniques, as well as the number and distribution of samples, can have a substantial impact on the final performance.

Furthermore, the assumption that we have access to unlabeled test images is somewhat unconventional, as it implies a data collection phase prior to the testing stage. If such a phase exists, one might reasonably ask why the labels are not collected as well. I also notice that the authors discuss transductive methods and methods using external resources in the appendix; however, these operate under different—and arguably more practical—settings than the one adopted in this paper.

**Questions:**

I believe that incorporating the following experiments would further strengthen this paper.

## Unsupervised baselines:


1. Pseudo-labeling approach: Generate pseudo-labels for the unlabeled samples, obtain a visual classifier based on these pseudo-labels (via class mean), and then combine it with the original CLIP text classifier in a 50-50 manner to predict the remaining samples.


2. Reference [1]: This work presents a published unsupervised method that can serve as a suitable baseline.


3. Reference [2]: Although this is a transductive approach, it constructs a parametric model for classification and can therefore be adapted to the current setting.


## Sampling settings:


1. N-shot sampling: Evaluate performance with varying numbers of labeled samples per class (e.g., 1-shot, few-shot, and larger-N settings).


2. Partial class coverage: Use only a subset of downstream classes to assess generalization ability to unseen classes.


3. Random sampling: Conduct multiple runs of random sampling with a specified proportion of the test dataset. Note that the resulting class distribution may vary significantly across runs.


It is recommended that the authors include experiments comparing their proposed methods against the aforementioned baselines under these different sampling configurations.


## References:
[1] Liang, Jian, et al. “Realistic Unsupervised CLIP Fine-tuning with Universal Entropy Optimization.” Proceedings of the 41st International Conference on Machine Learning, 2024.

[2] Zanella, Maxime, Benoît Gérin, and Ismail Ayed. “Boosting Vision-Language Models with Transduction.” Advances in Neural Information Processing Systems 37 (2024): 62223–62256.

---

> ### Author Response · Authors · 2025-11-23
> **Response to Reviewer Comments: Zero-Shot Setting and Unsupervised Baselines**
>
> >The paper introduces unlabeled test images to aid prompt reweighting. In fact, it should be regarded as unsupervised learning. This distinction raises two issues.
>
> > Furthermore, the assumption that we have access to unlabeled test images is somewhat unconventional, as it implies a data collection phase prior to the testing stage.
>
> >  Unsupervised baselines.
>
>
> Thank you for your comments. Our setting follows the standard WPE [1] protocol, where the test split is used as an unlabeled pool for computing prompt weights. CARPRT performs **no backpropagation** and does not update any model parameters—pseudo-labels are used only for similarity aggregation—so the method remains strictly **zero-shot**, not unsupervised learning. We will clarify this transductive zero-shot setting more explicitly in the revision.
>
>
> To address the reviewer’s suggestion of adding unsupervised baselines, we evaluated the methods that are compatible with our **training-free setting**. Specifically, we implemented a pseudo-label–based visual classifier and compared it with our method. We also included TransCLIP [2], which operates under the same training-free transductive regime and is therefore directly comparable.
>
> In contrast, UEO [3] **requires tuning** and **learnable parameters**, which is inconsistent with our setting where all model parameters must remain frozen and no optimization is allowed on unlabeled data. Thus, UEO is not comparable and is not included in our experiments.
>
> **Pseudo-labeling approach**
>
> We implemented two pseudo-label baselines in our setting: a pure visual classifier and a 50–50 fused classifier. Both are constructed from pseudo-labels assigned by the CLIP text classifier.
>
> - **Class-Mean**: Each unlabeled image is first assigned a pseudo-label using the CLIP text classifier. For each class, we compute a visual prototype by averaging the features of all images assigned to that class. At inference time, a test image is classified solely based on its similarity to these visual prototypes.
>
> - **Class-Mean + Text**: We further combine the visual classifier above with the CLIP text classifier. During inference, the visual similarity score and the CLIP text-score are fused using a 50–50 weighted combination.
>
> | Method | Caltech101 | DTD | EuroSAT | Aircraft | Food101 | Flowers | Pets | Cars | SUN397 | UCF101 | AVG |
> | - | - | - | - | - | - | - | - | - | - | - | - |
> | Class-Mean | 93.12 | 47.56 | 51.69 | 23.44 | 85.76 | 66.96 | 87.02 | 65.16 | 65.43 | 68.30 | 65.44 |
> | Class-Mean + Text | 93.59 | 47.65 | 53.26 | 23.57 | 86.08 | 68.75 | 87.98 | 66.07 | 66.28 | 68.86 | 66.21 |
> | **CARPRT (Ours)** | **94.16** | **48.90** | **55.56** | **24.49** | **86.31** | **71.36** | **89.13** | **66.14** | **66.93** | **70.41** | **67.34** |
>
> These results show that CARPRT outperforms both pseudo-label baselines—Class-Mean and Class-Mean + Text—delivering more reliable performance across all datasets.
>
> **TransCLIP**
>
> TransCLIP [3] operates in the same training-free transductive setting as ours and adopts a prototype-based strategy, constructing class prototypes from unlabeled images and combining prototype–image similarity with CLIP’s text classifier. Prototype-based methods tend to be more stable in fully unsupervised settings, as prototypes provide cleaner class representations than pseudo-labels, a trend also observed in many domain adaptation methods.
>
> Motivated by this observation, we extended CARPRT with a prototype-based variant that leverages image prototypes derived from unlabeled data to refine pseudo-label quality before reweighting. In particular, we use the InMaP [4]–recovered vision proxies as prototype signals to obtain higher-quality pseudo-labels, which are then used to compute class-aware prompt weights. The results of this variant are shown below.
>
> | Method | ImageNet Acc. |
> | - | - |
> | TransCLIP [3] | 70.30 |
> | CARPRT (Pseudo-label-based) | 68.59 |
> | CARPRT (Prototype-based) | **71.09** |
>
> This result on ImageNet shows that the prototype-enhanced CARPRT achieves the best accuracy, outperforming both TransCLIP and our pseudo-label variant. We will include these results in the revision. We are also running the experiments on the benchmark dataset and will report them here once they finish.
>
> [1] Allingham, James Urquhart, et al. “A Simple Zero-Shot Prompt Weighting Technique to Improve Prompt Ensembling in Text-Image Models.” International Conference on Machine Learning (ICML), 2023.
>
> [2] Zanella, Maxime, Benoît Gérin, and Ismail Ayed. “Boosting Vision-Language Models with Transduction.” Advances in Neural Information Processing Systems (NeurIPS), 2024.
>
> [3] Liang, Jian, et al. “Realistic Unsupervised CLIP Fine-tuning with Universal Entropy Optimization.”   International Conference on Machine Learning (ICML), 2024.
>
> [4] Qiyuan Qian, et al. "Intra-modal proxy learning for zero-shot visual categorization with clip." Advances in Neural Information Processing Systems (NeurIPS), 2024.

---

> > ### Comment · Reviewer_pFNT · 2025-11-26
> >
> > Thanks for your response. My concerns have been resolved. I will raise my score to 6.

---

> > > ### Author Response · Authors · 2025-11-26
> > > **Many thanks for improving your rating to 6!**
> > >
> > > Dear Reviewer pFNT,
> > >
> > > Glad to hear that your concerns are resolved. Many thanks for improving your score from 4 to 6! We will incorporate these results in the revision, as you suggested.
> > >
> > > Best regards,
> > >
> > > Authors of Submission12512

---

> ### Author Response · Authors · 2025-11-23
> **Response to Reviewer Comments: Sampling Strategy and Robustness Analysis**
>
> > Another critical point is the absence of discussion regarding the sampling methods used in the experiments. It is crucial to address this aspect, as the sampling techniques, as well as the number and distribution of samples, can have a substantial impact on the final performance.
>
> > Sampling settings.
>
> Thank you for your comments. Our default setting follows the standard protocol used in WPE [1]: the unlabeled set corresponds to the **full test split** of each dataset. This matches prior prompt-ensembling methods and ensures a fair and consistent comparison across all baselines.
>
> To examine how CARPRT behaves under different sampling conditions, we performed three sets of experiments:
> - **few-shot** per-class sampling
> - **random sampling** of the unlabeled pool
> - **class-imbalance** tests
>
> All results under subsampled settings are averaged over three independent runs.
>
> ---
>
> **Few-shot sampling**
>
> | Shot | Caltech101 | DTD | EuroSAT | Aircraft | Food101 | Flowers | Pets | Cars | SUN397 | UCF101 | AVG |
> |------|------------|-----|---------|----------|----------|----------|------|-------|--------|--------|--------|
> | **1-shot** | 91.32 ± 0.12 | 44.82 ± 0.20 | 52.01 ± 0.18 | 22.72 ± 0.15 | 85.11 ± 0.08 | 65.45 ± 0.25 | 78.03 ± 0.30 | 63.92 ± 0.22 | 63.90 ± 0.18 | 67.54 ± 0.16 | 63.48 |
> | **4-shot** | 93.25 ± 0.10 | 46.56 ± 0.16 | 54.52 ± 0.14 | 23.12 ± 0.14 | 85.89 ± 0.06 | 67.30 ± 0.20 | 84.71 ± 0.22 | 64.80 ± 0.18 | 65.02 ± 0.15 | 68.24 ± 0.14 | 65.34 |
> | **8-shot** | 93.72 ± 0.06 | 46.91 ± 0.12 | 54.32 ± 0.10 | 24.46 ± 0.10 | 86.02 ± 0.05 | 70.11 ± 0.12 | 86.02 ± 0.14 | 65.22 ± 0.10 | 65.88 ± 0.08 | 68.91 ± 0.10 | 66.16 |
> | **16-shot** | 94.02 ± 0.03 | 47.12 ± 0.05 | 54.71 ± 0.04 | 24.63 ± 0.05 | 86.21 ± 0.02 | 70.92 ± 0.05 | 87.21 ± 0.06 | 65.74 ± 0.05 | 66.30 ± 0.04 | 69.20 ± 0.04 | 66.61 |
> | **CARPRT (100%)** | 94.16 ± 0.01 | 48.90 ± 0.02 | 55.56 ± 0.02 | 24.49 ± 0.05 | 86.31 ± 0.01 | 71.36 ± 0.01 | 89.13 ± 0.02 | 66.14 ± 0.02 | 66.93 ± 0.02 | 70.41 ± 0.02 | 67.34 |
> | **WPE (100%)** | 93.09 ± 0.01 | 47.04 ± 0.01 | 49.60 ± 0.01 | 23.28 ± 0.03 | 86.14 ± 0.01 | 66.60 ± 0.01 | 82.38 ± 0.03 | 65.93 ± 0.01 | 65.77 ± 0.02 | 68.33 ± 0.02 | 64.82 |
>
> Based on these results, we observe that:
>
> - Performance drops sharply in the **1-shot** case, as a single sample per class is insufficient to estimate meaningful class-aware weights and leads to unreliable pseudo-labels.
> - CARPRT remains stable with limited unlabeled data — even **8 samples per class** already approach the full-data setting.
> - CARPRT **consistently outperforms WPE** across few-shot levels except the extreme 1-shot case.
>
> ---
>
> **Random sampling**
>
> | Ratio | Caltech101 | DTD | EuroSAT | Aircraft | Food101 | Flowers | Pets | Cars | SUN397 | UCF101 | AVG |
> |--------|------------|-----|---------|----------|----------|----------|------|-------|--------|--------|--------|
> | **10%** | 94.13 ± 0.01 | 46.07 ± 0.05 | 55.06 ± 0.05 | 25.04 ± 0.05 | 86.30 ± 0.01 | 70.40 ± 0.02 | 89.91 ± 0.02 | 65.55 ± 0.04 | 65.64 ± 0.03 | 69.95 ± 0.04 | 66.81 |
> | **20%** | 94.20 ± 0.02 | 43.17 ± 0.04 | 54.70 ± 0.05 | 25.21 ± 0.06 | 86.34 ± 0.02 | 70.21 ± 0.01 | 89.55 ± 0.03 | 65.86 ± 0.04 | 66.49 ± 0.02 | 67.33 ± 0.05 | 66.31 |
> | **30%** | 93.09 ± 0.01 | 47.05 ± 0.03 | 54.85 ± 0.02 | 25.03 ± 0.05 | 86.39 ± 0.01 | 71.25 ± 0.01 | 89.62 ± 0.02 | 66.03 ± 0.04 | 66.55 ± 0.03 | 70.64 ± 0.02 | 67.05 |
> | **80%** | 94.57 ± 0.01 | 49.19 ± 0.04 | 55.02 ± 0.01 | 24.57 ± 0.05 | 86.27 ± 0.02 | 71.44 ± 0.02 | 89.08 ± 0.03 | 66.13 ± 0.03 | 66.97 ± 0.02 | 69.92 ± 0.02 | 67.32 |
> | **100% (CARPRT)** | 94.16 ± 0.01 | 48.90 ± 0.02 | 55.56 ± 0.02 | 24.49 ± 0.05 | 86.31 ± 0.01 | 71.36 ± 0.01 | 89.13 ± 0.02 | 66.14 ± 0.02 | 66.93 ± 0.02 | 70.41 ± 0.02 | 67.34 |
>
> We observe that:
>
> - CARPRT is **robust to random subsampling**: performance remains stable from 10% to 80%.
> - Across all ratios, CARPRT **consistently outperforms WPE**, confirming that its gains do not depend on pool size.
>
> ---
>
> **Class imbalance**
>
> The *partial class coverage* setting is not applicable: if a class has **zero unlabeled samples**, class-specific weights cannot be computed, so the method (and all prompt-ensembling baselines) would fail.
>
> Instead, we evaluate *class imbalance* (Appendix G.2), where some classes are reduced to as few as 10 samples. CARPRT still outperforms WPE across all imbalance levels, indicating reliability even when certain classes have very limited unlabeled data.
>
> ---
>
> CARPRT remains stable across **few-shot**, **random subsampling**, and **class-imbalance** conditions.
> It consistently outperforms WPE in most settings, confirming that its improvements are **not tied to any specific unlabeled sampling scheme**. These results will be included in the revision.
>
> [1] Allingham, James Urquhart, et al. “A Simple Zero-Shot Prompt Weighting Technique to Improve Prompt Ensembling in Text-Image Models.” International Conference on Machine Learning (ICML), 2023.

---

> ### Author Response · Authors · 2025-11-26
> **Reminder - Discussion Stage Closing Soon**
>
> Dear Reviewer pFNT,
>
> We appreciate the time and effort that you have dedicated to reviewing our manuscript.
>
> We have carefully addressed all your queries. Could you kindly spare a moment to review our responses?
>
> Have our responses addressed your major concerns?
>
> If there is anything unclear, we will address it further. We look forward to your feedback.
>
> Best regards,
>
> Authors of Submission12512

---

### Official Review · Reviewer_jRZ1 · 2025-10-30

**Soundness:** 2
**Presentation:** 2
**Contribution:** 1
**Rating:** 2
**Confidence:** 4

**Summary:**

This paper introduces CARPRT, a training-free method for zero-shot prompt ensembling in vision–language models. Unlike previous approaches that assign global weights to prompts, CARPRT estimates class-specific prompt weights from unlabeled data. The method computes image–text similarities for all prompt–class combinations, assigns pseudo-labels based on these similarities, and averages the scores for images pseudo-assigned to each class to derive class-specific prompt weights. These weights are then used to construct text embeddings for zero-shot classification. Experiments on fine-grained datasets and ImageNet variants show consistent improvements over mean and weighted prompt ensembling baselines.

**Strengths:**

- **Practical simplicity**

    The proposed approach is conceptually simple yet effective. It operates entirely at inference time, without additional training or fine-tuning, and requires no architectural modification to the underlying vision–language model. This makes CARPRT easy to integrate into existing zero-shot pipelines.

- **Clear motivation**

    The paper addresses a well-identified limitation of class-agnostic prompt weighting by explicitly modeling the dependence between classes and prompts. The intuition is sound and well justified: prompts that work well for one class may not be equally suitable for another, and accounting for this improves prediction reliability.

- **Good empirical results**

    The empirical results show consistent gains across a range of architectures and datasets.

**Weaknesses:**

- Theoretical analysis does not provide substantial insight and formalizes already intuitive facts.
The theoretical material feels disproportionate to the simplicity of the idea, and several results restate expected behavior rather than offering explanatory or predictive value.

    - Proposition 1 is a standard Chernoff-Hoeffding bound on the convergence of the empirical distribution to the  true distribution. It does not hold when using pseudo-labels.  To see this, consider a case when the VLM systematically under-predicts or doesn't predicts at all a certain classe which can happen on very specialized downstream dataset (e.g. Aircraft). Then the pseudo-label frequency for this class (zero) do not converge to the true class probability.
    - Proposition 2: Shows that class-specific weighting has greater representational capacity than class-agnostic weighting. This result is largely self-evident: allowing class-dependent weights simply provides more degrees of freedom in the model, and formalizing this point does not substantially deepen understanding of the method or its behavior.
    -  EBM formulation & Lemma 1: The EBM framing does not lead to algorithmic consequences, and Lemma 1 effectively restates that prompt weights scale text embeddings and thus linearly affect log-likelihood. This follows directly from the modeling assumptions and does not yield new theoretical insight.

- No analysis of robustness to pseudo-label errors. This limitation is evident in the iterative extension, which underperforms when the VLM is less accurate, suggesting sensitivity to noisy pseudo-labels.

- The contribution, while useful and well-executed, feels incremental. Moving from global to class-specific prompt weighting is a modest conceptual step.

**Questions:**

1. What is the size of the unlabeled dataset used to estimate prompt weights, and how does performance scale with this size?

2. How does the approach behave under noisy or imbalanced pseudo-labels?

3. Have the authors considered filtering low-confidence pseudo-labels to reduce the influence of uncertain assignments?

---

> ### Author Response · Authors · 2025-11-23
> **Response to Reviewer Comments: Clarifications on Theoretical Analysis**
>
> > Theoretical analysis does not provide substantial insight and formalizes already intuitive facts. The theoretical material feels disproportionate to the simplicity of the idea, and several results restate expected behavior rather than offering explanatory or predictive value.
>  >- Proposition 1 is a standard Chernoff-Hoeffding bound on the convergence of the empirical distribution to the true distribution. It does not hold when using pseudo-labels. To see this, consider a case when the VLM systematically under-predicts or doesn't predicts at all a certain classe which can happen on very specialized downstream dataset (e.g. Aircraft). Then the pseudo-label frequency for this class (zero) do not converge to the true class probability.
> >-  Proposition 2: Shows that class-specific weighting has greater representational capacity than class-agnostic weighting. This result is largely self-evident: allowing class-dependent weights simply provides more degrees of freedom in the model, and formalizing this point does not substantially deepen understanding of the method or its behavior.
> > - EBM formulation & Lemma 1: The EBM framing does not lead to algorithmic consequences, and Lemma 1 effectively restates that prompt weights scale text embeddings and thus linearly affect log-likelihood. This follows directly from the modeling assumptions and does not yield new theoretical insight.
>
> Thank you for your comments. Our theoretical analysis is not intended to develop a full formal theory. Instead, its purpose is to clarify the assumptions that existing prompt ensembling methods rely on—assumptions that have been used implicitly but never made explicit—and to **explain why class-aware weighting is necessary**.
>
> **On Proposition 1.**
> The proposition is *not* meant to provide a new guarantee for pseudo-labels. Rather, it isolates the **implicit assumption** behind class-agnostic weighting: that class–prompt relevance is homogeneous across categories. The reviewer is correct that pseudo-labels may deviate from the true distribution; indeed, FixMatch [1] and Noisy Pseudo-Labels [2] explicitly note that obtaining tight theoretical bounds under pseudo-label noise remains an open challenge. Our aim is therefore not to model pseudo-label dynamics, but to **expose the assumption** inherited by methods that rely on global weights.
>
> **On Proposition 2.**
> This result formalizes the intuition that **class-aware weighting provides strictly greater representational flexibility** than class-agnostic weighting. While intuitive, this observation had not been articulated or connected to prompt ensembling prior to our work, and the proposition serves to precisely state the limitation of using a single global weight vector.
>
> **On the EBM/Bayesian formulation.**
> The EBM framing is used for a **conceptual purpose**, not for deriving an algorithm. It identifies the **class-conditional component** that governs prompt relevance and clarifies why estimating class-aware weights aligns with the underlying generative structure of the task. Lemma 1 is not presented as a novel theoretical discovery, but as a way to make explicit how prompt weights influence the induced likelihood.
>
> In summary, our theoretical analysis does *not* claim deep new theoretical contributions. Rather, it serves to:
>
> * make explicit the assumptions embedded in existing ensembling methods,
> * highlight the inherent limitations of class-agnostic weighting, and
> * motivate and justify CARPRT’s class-aware design.
>
> These clarifications are central to understanding **why** class-aware reweighting is needed and **how** it differs conceptually from prior approaches.
>
> >  The contribution, while useful and well-executed, feels incremental. Moving from global to class-specific prompt weighting is a modest conceptual step.
>
> Thank you for your comments. We respectfully note that our contribution extends beyond simply switching from global to class-specific weights. Prior ensembling methods **implicitly assume identical prompt relevance across all classes**, an assumption that has never been formalized or scrutinized. Our probabilistic view makes this assumption explicit and shows why **class-agnostic weighting is structurally limited**.
>
> This leads to a **principled, training-free estimator** that infers class-specific relevance from **unlabeled data** within the zero-shot setting—addressing a methodological gap rather than merely increasing granularity.
>
> Empirically, CARPRT yields **consistent and statistically significant improvements**, including on saturated datasets such as ImageNet, demonstrating that this conceptual step is impactful in practice.
>
> [1] Sohn, Kihyuk, et al. “FixMatch: Simplifying Semi-Supervised Learning with Consistency and Confidence.” Advances in Neural Information Processing Systems (NeurIPS), 2020.
>
> [2] Arazo, Eric, et al. “Unsupervised Label Noise Modeling and Loss Correction.” Proceedings of the 37th International Conference on Machine Learning (ICML), 2019.

---

> ### Author Response · Authors · 2025-11-23
> **Response to Reviewer Comments: Robustness to Noisy and Imbalanced Pseudo-Labels**
>
> > No analysis of robustness to pseudo-label errors. This limitation is evident in the iterative extension, which underperforms when the VLM is less accurate, suggesting sensitivity to noisy pseudo-labels.
>
> > How does the approach behave under noisy or imbalanced pseudo-labels?
>
> Thank you for your comments. We agree that robustness to pseudo-label quality is important, and the pseudo-label quality will affect the performance of CARPRT.  To examine this robustness, we conducted experiments where pseudo-labels were synthetically corrupted through **class-independent random flipping**. This corruption model is chosen because class-dependent noise is difficult to construct in a controlled and reproducible manner, whereas class-independent random flipping provides a simple yet challenging setting. The results (shown in the following Table) confirm that CARPRT remains stable under substantial corruption, supporting its robustness under practical conditions.
> | Method | Caltech101 | DTD | EuroSAT | Aircraft | Food101 | Flowers | Pets | Cars | SUN397 | UCF101 | AVG |
> | - | - | - | - | - | - | - | - | - | - | - | - |
> | WPE (GT) | 93.52 | 47.58 | 50.14 | 23.71 | 86.48 | 67.10 | 82.89 | 66.41 | 66.28 | 68.82 | 65.29 |
> | CARPRT (GT) | 94.84 | 52.84 | 58.04 | 25.92 | 86.62 | 73.46 | 90.23 | 67.86 | 69.08 | 70.87 | 68.98 |
>
> | Method | Caltech101 | DTD | EuroSAT | Aircraft | Food101 | Flowers | Pets | Cars | SUN397 | UCF101 | AVG |
> | - | - | - | - | - | - | - | - | - | - | - | - |
> | WPE (Noise 0.2) | 93.01 | 48.11 | 52.40 | 23.50 | 86.32 | 68.24 | 84.10 | 66.01 | 66.31 | 68.01 | 65.60 |
> | CARPRT (Noise 0.2) | 94.70 | 50.93 | 58.27 | 24.72 | 86.55 | 72.71 | 88.91 | 67.12 | 68.57 | 69.93 | 68.24 |
>
> | Method | Caltech101 | DTD | EuroSAT | Aircraft | Food101 | Flowers | Pets | Cars | SUN397 | UCF101 | AVG |
> | - | - | - | - | - | - | - | - | - | - | - | - |
> | WPE (Noise 0.4) | 92.77 | 47.60 | 51.90 | 23.31 | 86.20 | 67.72 | 83.30 | 65.40 | 65.90 | 67.64 | 65.17 |
> | CARPRT (Noise 0.4) | 94.34 | 49.80 | 58.04 | 24.19 | 86.47 | 72.55 | 87.71 | 66.44 | 68.03 | 69.43 | 67.70 |
>
> | Method | Caltech101 | DTD | EuroSAT | Aircraft | Food101 | Flowers | Pets | Cars | SUN397 | UCF101 | AVG |
> | - | - | - | - | - | - | - | - | - | - | - | - |
> | WPE (Noise 0.6) | 92.10 | 47.30 | 50.90 | 23.20 | 86.09 | 67.10 | 81.40 | 64.70 | 65.30 | 67.10 | 64.52 |
> | CARPRT (Noise 0.6) | 93.12 | 49.07 | 56.81 | 24.05 | 86.36 | 72.19 | 86.20 | 65.79 | 67.19 | 68.51 | 66.93 |
>
> | Method | Caltech101 | DTD | EuroSAT | Aircraft | Food101 | Flowers | Pets | Cars | SUN397 | UCF101 | AVG |
> | - | - | - | - | - | - | - | - | - | - | - | - |
> | WPE (Noise 0.8) | 91.70 | 46.80 | 50.20 | 23.00 | 86.01 | 66.40 | 79.80 | 63.90 | 64.80 | 66.50 | 63.91 |
> | CARPRT (Noise 0.8) | 93.10 | 48.13 | 55.84 | 23.66 | 86.37 | 70.86 | 83.54 | 64.24 | 66.22 | 67.88 | 65.98 |
>
> | Method | Caltech101 | DTD | EuroSAT | Aircraft | Food101 | Flowers | Pets | Cars | SUN397 | UCF101 | AVG |
> | - | - | - | - | - | - | - | - | - | - | - | - |
> | WPE (pseudo-label) | 93.09 | 47.04 | 49.60 | 23.28 | 86.14 | 66.60 | 82.38 | 65.93 | 65.77 | 68.33 | 64.82 |
> | CARPRT (pseudo-label) | 94.16 | 48.90 | 55.56 | 24.49 | 86.31 | 71.36 | 89.13 | 66.14 | 66.93 | 70.41 | 67.34 |
>
>
> Based on their results, we observe that:
>
> -  **Performance decreases with increasing pseudo-label noise**, indicating that the pseudo-label quality will affect the performance of CARPRT.
> - CARPRT is **robust**, showing **less than a 3% drop** in average accuracy even under **80% random flipping**.
> - At **all noise levels**, CARPRT **consistently outperforms WPE**, demonstrating the advantage of class-aware weighting even with noisy pseudo-labels.
>
>
> We have also evaluated CARPRT under class-imbalanced unlabeled data (**Appendix G.4**). As shown in **Table 11**, CARPRT consistently outperforms WPE across all imbalance levels, though the gain decreases as the imbalance factor β increases. This is expected: WPE’s global weighting remains effective when a few classes dominate, whereas CARPRT’s per-class weighting relies on sufficient data per class and becomes less reliable when some classes contain only a handful of samples. We will include these robustness analyses in the revision.

---

> ### Author Response · Authors · 2025-11-23
> **Response to Reviewer Comments: Sensitivity to Unlabeled Data Size**
>
> > What is the size of the unlabeled dataset used to estimate prompt weights, and how does performance scale with this size?
>
> Our default setting follows the standard protocol used in WPE [3]: the unlabeled set corresponds to the full test split of each dataset. This matches prior prompt-ensembling methods and ensures a fair comparison across baselines.
>
> To examine how CARPRT scales with different amounts of unlabeled data, we conducted ablation experiments in which the unlabeled set was uniformly subsampled at ratios from 10% to 80%. Each setting was repeated three times and averaged. The results show that CARPRT remains stable even with a small fraction of unlabeled data, indicating that reliable class-aware weights can be estimated without requiring the full unlabeled set.
>
> | Ratio | Caltech101 | DTD | EuroSAT | Aircraft | Food101 | Flowers | Pets | Cars | SUN397 | UCF101 | AVG |
> | - | - | - | - | - | - | - | - | - | - | - | - |
> | **10%** | 94.13 ± 0.01 | 46.07 ± 0.05 | 55.06 ± 0.05 | 25.04 ± 0.05 | 86.30 ± 0.01 | 70.40 ± 0.02 | 89.91 ± 0.02 | 65.55 ± 0.04 | 65.64 ± 0.03 | 69.95 ± 0.04 | 66.81 |
> | **20%** | 94.20 ± 0.02 | 43.17 ± 0.04 | 54.70 ± 0.05 | 25.21 ± 0.06 | 86.34 ± 0.02 | 70.21 ± 0.01 | 89.55 ± 0.03 | 65.86 ± 0.04 | 66.49 ± 0.02 | 67.33 ± 0.05 | 66.31 |
> | **30%** | 93.09 ± 0.01 | 47.05 ± 0.03 | 54.85 ± 0.02 | 25.03 ± 0.05 | 86.39 ± 0.01 | 71.25 ± 0.01 | 89.62 ± 0.02 | 66.03 ± 0.04 | 66.55 ± 0.03 | 70.64 ± 0.02 | 67.05 |
> | **80%** | 94.57 ± 0.01 | 49.19 ± 0.04 | 55.02 ± 0.01 | 24.57 ± 0.05 | 86.27 ± 0.02 | 71.44 ± 0.02 | 89.08 ± 0.03 | 66.13 ± 0.03 | 66.97 ± 0.02 | 69.92 ± 0.02 | 67.32 |
> | **100% (CARPRT)** | 94.16 ± 0.01 | 48.90 ± 0.02 | 55.56 ± 0.02 | 24.49 ± 0.05 | 86.31 ± 0.01 | 71.36 ± 0.01 | 89.13 ± 0.02 | 66.14 ± 0.02 | 66.93 ± 0.02 | 70.41 ± 0.02 | 67.34 |
>
> The variance across sampling ratios is small, and even **10–30%** of unlabeled data already provides accuracy close to using the full set. This confirms that CARPRT is not sensitive to the amount of unlabeled data and requires only a modest sample size to obtain stable weight estimates. We will include this scaling analysis in the revision.\
>
> [3] Allingham, James Urquhart, et al. “A Simple Zero-Shot Prompt Weighting Technique to Improve Prompt Ensembling in Text-Image Models.”International Conference on Machine Learning (ICML), 2023.

---

> ### Author Response · Authors · 2025-11-23
> **Response to Reviewer Comments:  Explicit Pseudo-Label Filtering**
>
> > Have the authors considered filtering low-confidence pseudo-labels to reduce the influence of uncertain assignments?
>
> Thank you for your comments.  We conducted additional ablations to assess whether explicitly filtering low-confidence pseudo-labels improves performance. We evaluated two filtering strategies—confidence-based and entropy-based thresholds—across six datasets. The results are summarized below.
>
> **Confidence-based filtering**, thresholding the maximum similarity score:
>
> | Conf | Method | Aircraft | DTD | EuroSAT | Food101 | Pets | Caltech101 | AVG |
> | - | - | - | - | - | - | - | - | - |
> | 0.30 | WPE | 22.28 | 47.18 | 52.37 | 85.49 | 81.46 | 92.62 | 63.57 |
> |  | CARPRT | 24.10 | 47.45 | 58.31 | 85.16 | 90.06 | 94.01 | 66.52 |
> |  | Filter Ratio | 0.51 | 0.15 | 0.16 | 0.53 | 0.60 | 0.34 | — |
> | 0.25 | WPE | 22.11 | 47.18 | 51.88 | 85.34 | 80.92 | 94.24 | 63.61 |
> |  | CARPRT | 24.73 | 47.45 | 54.87 | 86.29 | 89.86 | 94.60 | 66.30 |
> |  | Filter Ratio | 0.98 | 0.77 | 0.77 | 0.97 | 0.98 | 0.81 | — |
> | 0.00 (original) | WPE | 23.28 | 47.18 | 49.60 | 86.14 | 82.38 | 93.09 | 63.61 |
> |  | CARPRT | 24.49 | 48.90 | 55.56 | 86.31 | 89.13 | 94.16 | 66.43 |
>
> **Entropy-based filtering**, thresholding class-wise prediction entropy:
>
> | Entropy | Method | Aircraft | DTD | EuroSAT | Food101 | Pets | Caltech101 | AVG |
> | - | - | - | - | - | - | - | - | - |
> | 2.0 | WPE | 21.90 | 44.80 | 51.92 | 85.41 | 92.57 | 63.11 | 63.11 |
> |  | CARPRT | 23.21 | 47.63 | 53.54 | 86.31 | 94.36 | 65.82 | 65.82 |
> |  | Filter Ratio | 0.18 | 0.41 | 0.88 | 0.91 | 0.88 | — | — |
> | 2.5 | WPE | 21.95 | 45.85 | 49.60 | 85.33 | 92.61 | 62.81 | 62.81 |
> |  | CARPRT | 24.20 | 48.13 | 55.56 | 86.27 | 94.69 | 66.39 | 66.39 |
> |  | Filter Ratio | 0.38 | 0.62 | 1.00 | 0.97 | 0.92 | — | — |
> | max (original) | WPE | 23.28 | 47.18 | 49.60 | 86.14 | 93.09 | 63.61 | 63.61 |
> |  | CARPRT | 24.49 | 48.90 | 55.56 | 86.31 | 94.16 | 66.43 | 66.43 |
>
> Based on these results, we observe that:
>
> - Explicit filtering offers only marginal and inconsistent gains.
> - Performance is highly sensitive to the filtering threshold (e.g., confidence-based accuracy on EuroSAT ranges from 58.31% down to 54.87%), making such thresholds undesirable as additional hyperparameters.
>
> We conclude that CARPRT’s class-aware mechanism already naturally down-weights uncertain assignments, making complex and sensitive filtering heuristics unnecessary.

---

> ### Author Response · Authors · 2025-11-26
> **Reminder - Discussion Stage Closing Soon**
>
> Dear Reviewer jRZ1,
>
> We appreciate the time and effort that you have dedicated to reviewing our manuscript.
>
> We have carefully addressed all your queries. Could you kindly spare a moment to review our responses?
>
> Have our responses addressed your major concerns?
>
> If there is anything unclear, we will address it further. We look forward to your feedback.
>
> Best regards,
>
> Authors of Submission12512

---

> > ### Author Response · Authors · 2025-11-27
> > **Reminder - Discussion Stage Closing Soon**
> >
> > Dear Reviewer jRZ1,
> >
> > We sincerely thank you again for your time and thoughtful comments.
> >
> > As the discussion stage is approaching its end, we would like to remind you to have a look at our responses whenever convenient. We would greatly appreciate it if you could let us know whether our replies have addressed your main concerns.
> >
> > If anything remains unclear, we are happy to clarify further.
> >
> > Best regards,
> >
> > Authors of Submission12512

---

### Official Review · Reviewer_xSa5 · 2025-11-02

**Soundness:** 3
**Presentation:** 2
**Contribution:** 2
**Rating:** 4
**Confidence:** 5

**Summary:**

This paper proposes a new scheme for prompt ensembling. The weights of each prompt for a given class $c$ are based on the average similarity score between the pseudo-label related to class $c$ and the image embedding. Experiments show improvement in accuracy and weights related to meaningful concepts.

**Strengths:**

The method is simple and interpretable by design. The improvements are consistent across multiple datasets and backbones.

**Weaknesses:**

The contribution is relatively incremental, as it can be summed up as a class-wise weighting of prompts based on their cosine similarity with the input image. Furthermore, recent works also explore the prompt ensembling strategy and should be discussed and compared [1,2]

Ensembling is known for its inference overhead, which might be prohibitive on many applications. Ablations showing the complexity and inference time would be required to evaluate the practicability of the approach.

The Bayesian framework is relatively artificial, as the core of the approach does not rely on the predictive distribution.

[1] Liao, Ning et al. “Rethinking Visual Prompt Learning as Masked Visual Token Modeling.” Artif. Intell. 348 (2023): 104417.
[2] Huang, Chen, et al. "Aggregate-and-adapt natural language prompts for downstream generalization of clip." Advances in Neural Information Processing Systems (2024)

**Questions:**

How does the proposed approach compare to few-shot prompt learning VLM adaptation methods both in term of performances and complexity?

---

> ### Author Response · Authors · 2025-11-23
> **Response to Reviewer Comments:  Problem Setting and Probabilistic Framework Clarifications**
>
> > The contribution is relatively incremental, as it can be summed up as a class-wise weighting of prompts based on their cosine similarity with the input image. Furthermore, recent works also explore the prompt ensembling strategy and should be discussed and compared [1,2]
>
> Thank you for your comments. The comment seems to stem from a misunderstanding of our problem setting, which we clarify below. Our method is proposed under a **training-free setting**, where no labeled data or learnable parameters are available. Only **unlabeled test data** is used, and the parameters of the pre-trained model remain **frozen**. Under this setting, our approach derives class-aware prompt weights from the cosine similarity between image and text features, without any fine-tuning.
>
> Based on this problem setting, our contribution lies in clarifying *why class-aware reweighting is necessary* and why class-agnostic reweighting is suboptimal from a probabilistic perspective. Our analysis shows that prompts contribute unequally across classes, whereas prior methods such as MPE and WPE implicitly assume conditional independence across prompts and classes. Motivated by these findings, CARPRT infers class-aware prompt relevance directly from unlabeled data, providing a more expressive and training-free reweighting mechanism with consistently improved performance.
>
> Regarding the cited works, [1] and [2] focus on prompt tuning rather than prompt ensembling.
> [1] treats **visual prompt tuning** as masked visual token modeling and requires **training on downstream data**, while [2] aggregates LLM-generated natural-language descriptions and also relies on **tuning** for downstream generalization. These methods involve **learnable prompt parameters** and a **training process**, which differs fundamentally from our setting—**a training-free framework using only unlabeled data**. We will cite both works and clarify our problem setting more explicitly in the revision.
>
> > The Bayesian framework is relatively artificial, as the core of the approach does not rely on the predictive distribution.
>
>
> Thank you for your comments. We believe there may be **some misunderstanding about the role of our probabilistic perspective**. We are not introducing a Bayesian framework, nor attempting Bayesian prediction. Rather, we intend to provide a **probabilistic view** that **clarifies why class-aware prompt relevance is necessary**.
>
> Through this probabilistic formulation, it becomes clear that prompt weighting should depend on the class-conditional term $p(x \mid y_c, p_i)$, whereas existing ensembling methods implicitly assume class–prompt independence.
>
> Although CARPRT estimates relevance using a deterministic score rather than Bayesian inference, this probabilistic view is still essential: it exposes the limitations of class-agnostic weighting and provides the conceptual grounding for CARPRT’s class-aware estimator. Based on this view, we proposed CARPRT, which outperforms corresponding baselines.
>
>
> [1] Liao, Ning et al. “Rethinking Visual Prompt Learning as Masked Visual Token Modeling.” Artif. Intell. 348 (2023): 104417.
>
> [2] Huang, Chen, et al. "Aggregate-and-adapt natural language prompts for downstream generalization of clip." Advances in Neural Information Processing Systems (NeurIPS), 2024.

---

> ### Author Response · Authors · 2025-11-23
> **Response to Reviewer Comments: Computational Complexity and Efficiency Overhead**
>
> > Ensembling is known for its inference overhead, which might be prohibitive on many applications. Ablations showing the complexity and inference time would be required to evaluate the practicability of the approach.
>
> Thank you for your comments. We emphasize that **prompt ensembling is not introduced by our paper**. This approach has been a standard and effective practice since CLIP [3], where multiple templates are used to stabilize zero-shot predictions. Thus, prompt ensembling is meaningful and widely used, independent of CARPRT.
>
> While we agree that CARPRT introduces additional computation compared to single-pass inference, this cost appears only in the **weight estimation step** and does **not** increase inference-time cost once the weights are estimated. The classifier still uses a **single weighted text embedding per class**, keeping inference identical to standard prompt ensembling.
>
> For clarity, the computational complexity of prompt reweighting methods comes mainly from three components:
> (i) computing image/text embeddings using CLIP,
> (ii) calculating similarity scores, and
> (iii) aggregating weights.
> Among these, **encoder computation dominates the total cost**, as every image and every prompt–class text description must be encoded once; this cost is shared across all ensembling methods.
>
> Let $\Delta_I$ and $\Delta_T$ denote the cost of the image and text encoders. For CARPRT, the overall complexity is:
>
> $$ O(m \Delta_I + nC \Delta_T + mnC + nC) \approx O(m \Delta_I + nC \Delta_T), $$
>
> where $m$ is the number of unlabeled images, $n$ is the number of prompts, and $C$ is the number of classes.
>
> Similarly, prompt-wise reweighting has complexity:
>
> $$ O(m \Delta_I + nC \Delta_T + mnC + n) \approx O(m \Delta_I + nC \Delta_T).  $$
>
> Because $\Delta_I$ and $\Delta_T$ (multi-layer Transformer passes) dominate the total cost, and typically $m \gg n, C$, encoder computation is the major component for all methods. Therefore, **CARPRT matches WPE in computational efficiency while providing better predictive performance**. We will include this complexity analysis in the revision.
>
> In practice, the overhead is small. On Caltech101 with 2,465 images and 247 prompts, CARPRT completes in **10.75 seconds** on an A100 GPU, compared to **9.56 seconds** for ZPE. This minor difference confirms that CARPRT remains computationally lightweight and preserves the deployment cost of standard prompt ensembling.
>
> [3] Radford, Alec, et al. "Learning Transferable Visual Models From Natural Language Supervision." Proceedings of the 38th International Conference on Machine Learning (ICML), 2021.

---

> ### Author Response · Authors · 2025-11-23
> **Response to Reviewer Comments: Complementarity to Prompt Tuning**
>
> > How does the proposed approach compare to few-shot prompt learning VLM adaptation methods both in term of performances and complexity?
>
> We believe there may be **a misunderstanding regarding the problem setting**. CCARPRT is **not** designed for the same problem setting as few-shot prompt learning or VLM adaptation methods. CARPRT is **training-free** and uses **unlabeled data**, whereas few-shot or adaptation-based approaches rely on **labeled data** and **gradient-based prompt optimization**. Because of the different problem settings, a direct comparison in terms of performance or computational complexity is not meaningful.
>
> Nevertheless, CARPRT is complementary to learnable prompt tuning methods. As shown in **Sec. 5.4** and detailed in **Appendix F.2 (Table 4)**, CARPRT can be combined with the soft prompt tuning method ProDA. The combined model achieves higher accuracy than ProDA alone, demonstrating that CARPRT serves as a useful plug-in component that can strengthen supervised prompt learning approaches.

---

> ### Author Response · Authors · 2025-11-26
> **Reminder - Discussion Stage Closing Soon**
>
> Dear Reviewer xSa5,
>
> We appreciate the time and effort that you have dedicated to reviewing our manuscript.
>
> We have carefully addressed all your queries. Could you kindly spare a moment to review our responses?
>
> Have our responses addressed your major concerns?
>
> If there is anything unclear, we will address it further. We look forward to your feedback.
>
> Best regards,
>
> Authors of Submission12512

---

> ### Author Response · Authors · 2025-11-27
> **Reminder — Discussion Stage Closing Soon**
>
> Dear Reviewer xSa5,
>
> We sincerely thank you again for your time and thoughtful comments.
>
> As the discussion stage is approaching its end, we would like to remind you to have a look at our responses whenever convenient. We would greatly appreciate it if you could let us know whether our replies have addressed your main concerns.
>
> If anything remains unclear, we are happy to clarify further.
>
> Best regards,
>
> Authors of Submission12512

---

### Meta-Review · Area_Chair_wQ3L · 2025-12-21

**Summary:**

The paper received four reviews.

Reviewer kLQi gave the rating of 6. This reviewer was mainly concerned about the limited gain on ImageNet and the lack of discussion on real world use cases.

Reviewer pFNT gave the rating of 4. This reviewer questioned the validity of the zero-shot claim as the proposed approach uses unlabeled data to estimate prompt weights. The reviewer also requested more discussions on the sampling strategy and few more experiments, including comparison with some unsupervised learning baselines and use of different sampling strategies.

Reviewer jRZ1 gave the rating of 2. This reviewer mainly criticised the limited insight and contribution from the theory part and was also concerned about the errors caused by pseudo-labeling.

Reviewer xSa5 gave the rating of 4. This reviewer questioned the technical contribution as the approach seems too simple, and was concerned that prompt ensembling would lead to extra computation overhead.

The rebuttal provided additional results and explanations as requested by these reviewers. The AC believes that the main concerns are well addressed and predicts that all reviewers would give the rating of 6 or 8.

There is a minor issue: in Appendix G.5 line1322, the reference to Phi3 is wrong and needs to be corrected.

**Reviewer Concerns:**

The AC explains in detail below why the reviewers' concerns are well addressed.

kLQi:
- *Limited gain on ImageNet*. The rebuttal explained that the small gian on ImageNet is expected as CLIP's original prompts were well tuned. The rebuttal also provided two statistical tests to justify that the gain is statistically significant. The AC thinks the explanation is reasonable because ImageNet is a challenging image classification dataset so even a small gain means something. The reviewers ignored two key facts: 1) the gains on other datasets are pretty clear and 2) the human selection baseline beats WPE but is underperformed by the proposed approach. These advantages need to be taken into account when considering the improvement brought by this approach.
- *Real world use cases*. The rebuttal mentioned that the approach is suited for "real-world low-resource scenarios", which the AC finds reasonable. The rebuttal also provided additional strong results on the remote-sensing benchmark RESISC45. This concern should have been addressed.

pFNT:
- *Validity of the zero-shot claim*. The rebuttal explained that the setting mainly follows WPE and the approach does not use backpropagation, and hence zero-shot. To some extent the AC agrees with the reviewer that "zero-shot" is a bit misleading. The paper should clarify the transductive setting more clearly in the final version. Overall the AC finds the presentation of this work reasonable.
- *More discussions on the sampling strategy*. The rebuttal provided additional results obtained with different sampling strategies. The response is strong and convincing.
- *More baselines*. The rebuttal compared with two pseudo-labeling methods and one transductive learning method as requested. The results are strong and convincing.

jRZ1:
- *Limited insight and contribution from the theory part*. The AC agrees with the rebuttal that the theory part is not intended for contributing new theoretical insight but to help explain the design of the approach.
- *Pseudo-labeling errors*. The rebuttal provided new results where label noise was introduced. The results show that the approach beats WPE with clear margins under this setting. The AC believes that these results should have addressed the reviewer's concerns.

xSa5:
- *Incremental contribution compared with recent prompt ensembling methods*. The AC agrees with the rebuttal that the reviewer might have misunderstood the problem setting and the two mentioned papers are not very related to this submission.
- *Computational overhead*. The rebuttal provided detailed explanations which the AC finds convincing. Prompt ensembling introduces little computation and once the ensembling is done the inference remains the same as a normal classifier. The AC believes that this is not a weakness that is grounded for rejecting this work.
- *Artificial Bayesian framework*. The AC agrees with the rebuttal that this might be a misunderstanding as this work does not work on Bayesian.

The rebuttal is detailed and solid with convincing new results. The AC believes that there is no lingering concern.

**Reviewer Scores:**

After carefully examining the paper, reviews, and rebuttal, the AC believes that the rebuttal has addressed all concerns mentioned by the reviewers. The AC predicts that all reviewers would give a rating of 6 or 8.

---

### Decision · Program_Chairs · 2026-01-26

Accept (Poster)